# Empirical Lagrangian parametrization for wind-driven mixing of buoyant particles at the ocean surface

Victor Onink[1,2,3], Erik van Sebille[3], and Charlotte Laufkötter[1,2]

[1]Climate and Environmental Physics, Physics Institute, University of Bern, Bern, Switzerland
[2]Oeschger Centre for Climate Change Research, University of Bern, Bern, Switzerland
[3]Institute for Marine and Atmospheric Research, Utrecht University, Utrecht, The Netherlands
**Correspondence:** Victor Onink (victor.onink@climate.unibe.ch)

**Abstract.** Turbulent mixing is a vital component of vertical particulate transport, but ocean global circulation models (OGCMs) generally have low resolution representations of near-surface mixing. Furthermore, turbulence data is often not provided in OGCM model output. We present 1D parametrizations of wind-driven turbulent mixing in the ocean surface mixed layer, which are designed to be easily included in 3D Lagrangian model experiments. Stochastic transport is computed by Markov-0 or Markov-1 models, and we discuss the advantages/disadvantages of two vertical profiles for the vertical diffusion coefficient $K_z$. All vertical diffusion profiles and stochastic transport models lead to stable concentration profiles for buoyant particles, which for particles with rise velocities of 0.03 and 0.003 m s$^{-1}$ agree relatively well with concentration profiles from field measurements of microplastics when Langmuir-circulation-driven turbulence is accounted for. Markov-0 models provide good model performance for integration timesteps of $\Delta t \approx 30$ seconds, and can be readily applied in studying the behaviour of buoyant particulates in the ocean. Markov-1 models do not consistently improve model performance relative to Markov-0 models, and require an additional parameter that is poorly constrained.

## 1 Introduction

Lagrangian models are essential tools to examine the transport of particulates in the ocean on a variety of spatial and temporal scales (Van Sebille et al., 2018), and have been used to study the movement of plastic particulates (Onink et al., 2019), oil (Samaras et al., 2014) and fish larvae (Paris et al., 2013). However, especially in the field of marine plastic modelling, most large scale modelling studies consider only virtual particles (henceforth referred to as particles) that float and remain at the ocean surface (Lebreton et al., 2018; Liubartseva et al., 2018; Onink et al., 2019, 2021), essentially simplifying the three dimensional ocean into a 2D system. While this does reduce the complexity of models, ultimately vertical transport processes need to be considered in order to have a complete understanding of oceanic particulate transport (Wichmann et al., 2019; Van Sebille et al., 2020).

In the case of buoyant particulates (particulates with a density lower than seawater), buoyancy is expected to return any particulates to the ocean surface. However, instead of all buoyant particulates accumulating at the ocean surface, both field

measurements (Kukulka et al., 2012; Kooi et al., 2016b) and regional large-eddy simulations (LES) model studies (e.g. Liang et al., 2012; Yang et al., 2014; Brunner et al., 2015; Taylor, 2018) indicate vertical concentration profiles throughout the mixed layer (ML). These profiles arise due to the balance between the particulate buoyancy and turbulent mixing flows, which are largely driven by wind and wave breaking at the ocean surface (Chamecki et al., 2019). While such profiles are commonly used to correct surface measurements of particulates such as microplastics (e.g. Law et al., 2014; Egger et al., 2020), it is difficult to

recreate such vertical mixing profiles in the ML outside of LES models, as vertical turbulent processes generally act on much smaller scales than is explicitly resolved in ocean global circulation models (OGCMs) (Taylor, 2018). In addition, while it is possible to represent mixing using the parametrization from Kukulka et al. (2012), this approach is only valid for depths up to several meters, while the mixed layer depth (MLD) can be hundreds of meters deep (Chamecki et al., 2019).

In this study we present numerical simulations of buoyant virtual particles in the ML with four 1D wind-driven mixing parametrizations. These mixing parametrizations have been specifically designed such that the code can be easily adapted to function within large-scale 3D Lagrangian models running with OGCM data, for cases where the vertical spatial scales might be too coarse to explicitly represent turbulent processes or where turbulence data might not be provided as model output. Using these parametrizations we calculate the vertical equilibrium profiles of buoyant particles within the ML as a function of

the particle rise velocities, the 10m wind speed and the MLD. Buoyant particles are found below the ML (Pieper et al., 2019; Choy et al., 2019; Egger et al., 2020), but diffusive mixing at such depths is likely not due to wind-driven turbulent mixing and therefore goes beyond the scope of this study. We test two methods for solving stochastic differential equations, and consider vertical diffusion coefficient profiles based on the KPP model (Large et al., 1994) and on Kukulka et al. (2012) extended by Poulain (2020). The modelled concentration profiles are then compared with measurements of vertical concentration profiles

of microplastics.

## 2  Model Framework

### 2.1  Lagrangian stochastic transport

Turbulence in the ocean occurs over a wide range of spatial and temporal scales, with Kolmogorov length and timescales of $\eta = (\nu^3/\epsilon)^{1/4} = 3 \times 10^{-4}$ m and $\tau_n = (\nu/\epsilon)^{1/2} = 0.1$ s (Landahl and Christensen, 1998) for turbulent kinetic energy $\epsilon = 10^{-4}$

m$^2$ s$^{-2}$ (Gaspar et al., 1990) and kinematic viscosity of seawater $\nu = 10^{-6}$ m$^2$ s$^{-1}$ (Riisgård and Larsen, 2007). The vertical resolution of OGCMs is typically on the order of meters and is therefore not capable of explicitly resolving all turbulent processes. Instead, turbulence due to sub-grid scale processes is generally represented stochastically. In our 1D vertical model, we simulate positively buoyant particles that are vertically transported due to stochastic turbulence and the particle rise velocity $w_{rise}$. For such particles, the particle trajectory $Z(t)$ can be computed with a stochastic differential equation (SDE) (Gräwe

 et al., 2012) as:

$$Z(t+dt) = Z(t) + (w_{rise} + \partial_z K_z)dt + \sqrt{2K_z}dW \tag{1}$$

$$Z(0) = 0 \tag{2}$$

where $K_z = K_z\big(Z(t)\big)$ is the vertical diffusion coefficient, $\partial_z K_z = \partial K_z/\partial z$, $dW$ is a Wiener increment with zero mean and variance $dt$ and we define the vertical axis $z$ as positive upward with $z = 0$ at the air–sea interface. The Euler-Maruyama (EM) scheme (Maruyama, 1955) is the simplest numerical approximation of equation 1, where infinitesimal terms $dt$ and $dW$ are replaced with the finite $\Delta t$ and $\Delta W$. Equation 1 can then be rewritten as (Gräwe et al., 2012):

$$w'(t) = \partial_z K_z + \frac{1}{\Delta t}\sqrt{2K_z}\Delta W \tag{3}$$

$$Z(t+\Delta t) = Z(t) + \big(w_{rise} + w'(t)\big)\Delta t \tag{4}$$

where $w'$ is the stochastic velocity perturbation due to turbulence. The turbulent transport has both a deterministic drift term and a stochastic term. This is the most basic form of representing turbulent particle transport, as turbulent perturbations on the particle position are assumed to be uncorrelated (Berloff and McWilliams, 2003). The drift term assures that the well-mixed condition is met, which states that an initially uniform particle distribution must remain uniform even with inhomogeneous turbulence (Brickman and Smith, 2002; Ross and Sharples, 2004). This approach, termed a Markov-0 (M-0) or random walk model, assumes that turbulent fluctuations exhibit no autocorrelation on timescales $\Delta t$, which for global-scale Lagrangian simulations can range from 30 seconds (Lobelle et al., 2021) to 30 minutes (Onink et al., 2019). However, measurements from Lagrangian ocean floats show this is an oversimplification, as coherent oceanic flow structures can induce velocity autocorrelations that can persist for significantly longer timescales (Denman and Gargett, 1983; Brickman and Smith, 2002).

A higher order approach is the Markov-1 (M-1) model, which assumes a degree of autocorrelation of particle velocities set by the Lagrangian integral timescale $T_L$. The turbulent velocity perturbation is now expressed as a Langevin equation, and with an EM numerical scheme the particle trajectory $Z(t)$ is computed as (Mofakham and Ahmadi, 2020):

$$Z(t+\Delta t) = Z(t) + \big(w_{rise} + w'(t)\big)\Delta t \tag{5}$$

$$w'(t+\Delta t) = \alpha w'(t) + \partial_z \sigma_w^2 \Delta t + \sqrt{\frac{2(1-\alpha)\sigma_w^2}{\Delta t}}\Delta W \tag{6}$$

where $\alpha = 1 - \Delta t/T_L$ and $\sigma_w^2 = \sigma_w^2(z,t)$ is the variance of $w'$, and we assume $\Delta t \leq T_L$. The influence of the initial turbulent fluctuations on subsequent fluctuations is set by $\alpha$, which in turn depends on the ratio between the integration timestep $\Delta t$ and $T_L$. However, empirical and theoretical estimates for $T_L$ range from 6-7 seconds (Kukulka and Veron, 2019) to 15-30 minutes (Denman and Gargett, 1983), and $T_L$ can also be depth dependent (Brickman and Smith, 2002). In large-eddy simulation (LES) models, $T_L = 4e/3C_0\epsilon$ where $e$ is the sub-grid scale turbulent kinetic energy, $C_0$ is a model constant determining diffusion in the velocity space and $\epsilon$ is the turbulent kinetic energy dissipation rate (Kukulka and Veron, 2019), but $e$ and $\epsilon$ are not commonly available variables in the output of OGCMs. However, it does indicate why model $T_L$ estimates vary widely, as $T_L$ describes

the autocorrelation of the particle velocity from its initial velocity due to unresolved sub-grid processes, which depends on the model resolution and setup in a given study. Since there is not a clear indication of the true value of $T_L$, we consider a range of values $\alpha \in [0, 0.1, 0.3, 0.5, 0.7, 0.95]$, corresponding to $T_L \in [1, 1.1, 1.4, 2, 3.3, 20] \times \Delta t$. As the depth dependence of $T_L$ is uncertain, we make the simplification that $\partial_z T_L = \partial_z \alpha = 0$. Since $\Delta t \leq T_L$, we use $K_z = \sigma_w^2 \Delta t$ (Brickman and Smith, 2002), which means that equation 6 becomes:

$$w'(t) = \alpha w'(t) + \partial_z K_z + \frac{1}{dt} \sqrt{2(1-\alpha) K_z} \Delta W \tag{7}$$

In this form, it is clear that equation 7 is equivalent to equation 4 when $\alpha = 0$. This is because when $\alpha = 0$, velocity perturbations $w'$ are assumed to be uncorrelated over timescales $\geq \Delta t$, which is equivalent to the M-0 formulation. M-1 stochastic models generally should lead to improved representation of diffusion in Lagrangian models (Berloff and McWilliams, 2003; Van Sebille et al., 2018), but it does require insight into turbulence statistics that have not yet been extensively studied in Lagrangian settings. For that reason, while even higher order Markov models are theoretically possible (Berloff and McWilliams, 2003), we limit this study to just the M-0 and M-1 approaches.

All Lagrangian simulations are run using Parcels v2.2.1 (Delandmeter and Sebille, 2019), which has been used for 1D, 2D and 3D particle oceanographic simulations (Fischer et al., 2021; Onink et al., 2021; Lobelle et al., 2021). The simulations start with 100,000 particles released at $Z(0) = 0$ and run for 12 hours. The model is one dimensional with horizontal velocities set to zero. The time-invariant vertical diffusion profiles are calculated with a 0.1 m vertical resolution, where the $K_z$ value at the exact particle location is linearly interpolated from these profiles. The vertical transport is calculated according to Equations 3 and 4 for M-0 simulations, and Equations 5 and 7 for M-1 simulations. We take $\Delta t = 30$ seconds, where the integration timestep is a compromise between accounting for turbulent transport on short timescales and computational cost for when the 1D model is integrated into a larger 3D Lagrangian model. We consider high, medium and low buoyancy particles with rise velocities of $w_{rise} \in [0.03, 0.003, 0.0003]$ m s$^{-1}$, which for plastic polyethylene ($\rho = 980$ kg m$^{-3}$) particles corresponds to spherical particles with diameters of 2.2, 0.4 and 0.1 mm (Enders et al., 2015). However, these particle sizes are rough indications of approximate particle sizes, as the buoyancy of particle depends on a combination of the particle size, shape, polymer density and degree of biofouling (Kooi et al., 2016b; Brignac et al., 2019; Kaiser et al., 2017). Relative to peak stochastic velocity perturbations $w'$ calculated from the vertical diffusion coefficients described in Section 2.2, the rise velocity of the high buoyancy particles dominate $w'$ except for the highest wind speeds, while turbulence dominates buoyancy for the medium and low buoyancy particles for almost all wind conditions (Table A1). The surface wind stress is computed from $u_{10} \in [0.85, 2.4, 4.35, 6.65, 9.3]$ m s$^{-1}$. The model domain is $z \in [-100, 0]$m, where we apply a ceiling boundary condition (BC) in which particles that cross the surface boundary are placed at $z = 0$. This BC assures that neither buoyancy or turbulence can transport particles out of the water column. Vertical concentration profiles are computed by binning the final particle locations into 0.5 m bins, and the concentrations are then normalized by the total number of particles in the simulation. The variability of the profiles at each depth level is calculated as the standard deviation over the final hour of each simulation.

## 2.2 Vertical diffusion profiles

Two vertical diffusion coefficient profiles are used, with the first based on Kukulka et al. (2012) and Poulain (2020). Kukulka et al. (2012) parametrized the near-surface vertical diffusion coefficient $K_z^S$ due to breaking waves as:

$$K_z^S = 1.5 u_{*w} \kappa H_s \tag{8}$$

for $z > -1.5 H_s$, where $\kappa = 0.4$ is the von Karman constant, $H_s$ is the significant wave height and $u_{*w}$ is the friction velocity of water. The significant wave height $H_s$ is parametrized as $H_s = 0.96 g^{-1} \beta_*^{3/2} u_{*a}^2$, where $g = 9.81$ m s$^{-2}$ is the accelation of gravity, $\beta_* = c_p / u_{*a}$ is the wave age, $c_p$ being the characteristic phase speed of the surface waves and $u_{*a} = \tau / \rho_a$ is the friction velocity of water. The friction velocity of air is based on the air density $\rho_a = 1.22$ kg m$^{-3}$ and the surface wind stress $\tau = C_D \rho_a u_{10}^2$, where $u_{10}$ is the 10m wind speed and $C_D$ is the drag coefficient (Large and Pond, 1981). Similarly, $u_{*w} = \tau / \rho_w$ with the seawater density $\rho_w = 1027$ kg m$^{-3}$. Following Kukulka et al. (2012), we assume a fully developed sea-state with $\beta_* = 35$. The Kukulka et al. (2012) parametrization is valid only for $z \approx -1.5 H_s$, and we extend the parametrization for greater depths using the eddy viscosity profile $\nu_z$ as found for oscillating grid turbulence by Poulain (2020):

$$\nu_z = \begin{cases} \nu^S \text{ if } z > -\gamma H_s \\ \nu^S (\gamma H_s)^{3/2} |z|^{-3/2} \text{ if } z < -\gamma H_s \end{cases} \tag{9}$$

where $\nu^S$ is the near surface eddy viscosity and $\gamma = 1.0$ is a multiple of $H_s$ that sets the depth to which $\nu^S$ is constant. This approach agrees with Kukulka et al. (2012) in predicting constant mixing for $z > -H_s$, where the eddy viscosity then drops proportional to $z^{-3/2}$ for greater depths. Oscillating grid turbulence (OGT) experiments are commonly used to study wave and wind induced turbulence (Fernando, 1991). As OGT experiments have been shown to reproduce turbulence decay laws of velocities and dissipation rates observed in the ocean ML (Thompson and Turner, 1975; Hopfinger and Toly, 1976; Craig and Banner, 1994), this provides some confidence in the modelling of the decay of near-surface eddy viscosity, although direct validation with field measurements of eddy viscosity have yet to occur. The diffusion coefficient $K_z$ depends on $\nu_z$ as $K_z = \nu_z / Sc_t$, where $Sc_t$ is the turbulent Schmidt number, and assuming $\partial_z Sc_t = 0$, combining equations 8 and 9 results in:

$$K_z = \begin{cases} K_z^S + K_B = 1.5 u_{*w} \kappa H_s + K_B \text{ if } z > -\gamma H_s \\ K_z^S (\gamma H_s)^{3/2} |z|^{-3/2} + K_B = 1.5 u_{*w} \kappa \gamma^{3/2} H_s^{5/2} |z|^{-3/2} + K_B \text{ if } z < -\gamma H_s \end{cases} \tag{10}$$

where $K_B = 3 \times 10^{-5}$ m$^2$ s$^{-1}$ is the dianeutral diffusion below the MLD (Waterhouse et al., 2014). The diffusion is thus constant for $z > -\gamma H_s$, below which $K_z \propto |z|^{-3/2}$, while the magnitude of $K_z$ increases for higher wind speeds (Fig. 1). Poulain (2020) implies $\gamma = 1.0$ while Kukulka et al. (2012) estimates $\gamma \approx 1.5$, so to test the model sensitivity we consider $\gamma \in [0.5, 1.0, 1.5, 2.0]$ (Figure 1). As $z \to -\infty$, $|z|^{-3/2} \to 0$, and therefore we include the bulk dianeutral diffusion $K_B$ to account for vertical mixing at depths below the influence of surface wave-driven turbulence. As both Kukulka et al. (2012) and Poulain et al. (2019) considered turbulence generated by breaking surface waves, we refer to this diffusion approach as Surface Wave Breaking (SWB) diffusion.

The second vertical diffusion coefficient profile is a local form of the K-profile parameterization (KPP) (Large et al., 1994; Boufadel et al., 2020), where $K_z$ is given by:

$$K_z = \left(\frac{\kappa u_{*w}}{\phi}\theta\right)(|z| + z_0)\left(1 - \frac{|z|}{MLD}\right) + K_B \tag{11}$$

where $\phi = 0.9$ is the "stability function" of the Monin-Obukov boundary layer theory, $\theta$ is a Langmuir circulation (LC) enhancement factor, and $z_0$ is the roughness scale of turbulence. As such, $K_Z$ rises from a small non-zero value at $z = 0$ to a maxima at $z = 1/3MLD$, before dropping to $K_z = K_B$ for $z \leq MLD$ (Fig. 1). In the original KPP formulation $K_z(z \leq MLD) = 0$ since the theory only applies to the surface mixed layer, so we add the same bulk dianeutral diffusion term $K_B$ as with the SWB profile (equation 10). Boufadel et al. (2020) examined a case where LC-driven turbulence was considered negligible and so $\theta = 1.0$. However, the presence of LC can increase turbulent mixing by a factor $\theta = 3 - 4$ (McWilliams and Sullivan, 2000) and has been shown to strongly affect the vertical concentration profiles of buoyant microplastic particles in LES experiments (Brunner et al., 2015; Kukulka and Brunner, 2015). Therefore, we examine $\theta \in [1.0, 2.0, 3.0, 4.0, 5.0]$. The roughness scale $z_0$, which can represent the surface roughness due to surface waves, depends on the wind speed and the wave age (Zhao and Li, 2019), and following Kukulka et al. (2012) we consider a wave age $\beta_* = c_p/u_{*a} = 35$ that is equivalent to $\beta = c_p/u_{10} = 1.21$. According to Zhao and Li (2019), the roughness scale is given by:

$$z_0 = 3.5153 \times 10^{-5}\beta^{-0.42}u_{10}^2/g \tag{12}$$

For $w_{10} = 0.85 - 9.30$ m s$^{-1}$, this means $z_0 = 2.38 \times 10^{-6} - 2.86 \times 10^{-4}$ m. To test the model sensitivity to $z_0$, we also consider an alternative scenario where $z_0 = 0.1 \times H_s = 1.76 \times 10^{-3} - 2.10 \times 10^{-1}$ m, following the same formulation $H_s = 0.96g^{-1}\beta_*^{3/2}u_{*a}^2$ as in Kukulka et al. (2012). This increases $K_z$ for $z \approx 0$, but does not significantly affect the magnitude $K_z$ at greater depths (Figure B1). The original KPP theory does not explicitly account for surface wave breaking, which would lead to larger non-zero $K_z$ at $z = 0$. While we do not claim that setting $z_0 = 0.1 \times H_s$ means that our KPP profile accounts for surface wave breaking turbulent mixing, it allows us to investigate the influence higher near-surface mixing would have on the modelled vertical concentration profiles. The MLD is the maximum depth of the surface ocean boundary layer formed due to interaction with the atmosphere, and in KPP theory the MLD is defined as the depth where the bulk Richardson number $Ri_B$ is first equal to a critical value $Ri_{crit}$. In the original formulation $Ri_{crit} = 0.3$ (Large et al., 1994), but $Ri_B$ can be difficult to compute in the field as this requires data for both vertical density and velocity shear profiles. In this study we prescribe MLD$= 20$ m, as this falls within the range of the MLD for field data used to evaluate the model (see Section 2.3).

## 2.3   Field data

We compiled a dataset of vertical plastic concentration profiles collected within the surface mixing layer to validate the modelled concentration profiles (Table 1), with a total of 90 profiles with 741 data points. Only Kooi et al. (2016b) directly measured the rise velocity of a subsample of the collected microplastic particulates, and showed that these particles were positively buoyant. However, the presence of all the other sampled particulates near the open ocean surface indicates they are unlikely to be

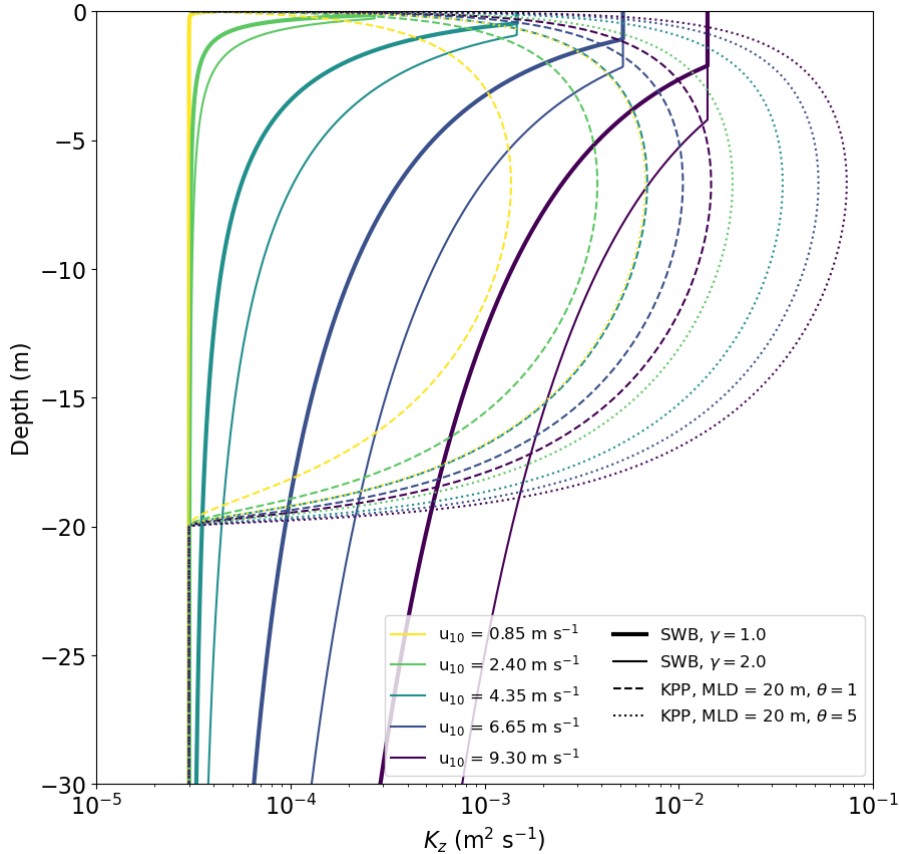

**Figure 1.** Vertical diffusion coefficient profiles for SWB and KPP diffusion under varying wind conditions. The KPP diffusion profile is calculated with $z_0$ according to Equation 12.

negatively buoyant. For all stations the wind speed was recorded and the MLD was determined from CTD data based on a temperature threshold (de Boyer Montégut et al., 2004). The majority of samples were collected in the North Atlantic (Kukulka et al., 2012; Kooi et al., 2016b; Pieper et al., 2019), and in regions with a relatively shallow MLD. Since wind-driven turbulent mixing isn't expected to influence the concentration depth profile below the MLD, we don't consider any measurements collected below 73 m. Measurements were collected with surface wind speeds up to $10.7$ m s$^{-1}$, with the majority of sampled concentrations being collected for $u_{10} = 3.4 - 7.9$ m s$^{-1}$ (535/741 data points).

Almost all measurements were collected with neuston nets, either multi-level nets simultaneously sampling fixed depth intervals (Kooi et al., 2016b) or using multi-stage nets that consecutively sample fixed depths or depth ranges (Kukulka et al. (2012); Egger et al. (2020); Amaral-Zettler (unpublished data)). These nets have mesh-sizes of $0.33$ mm, and will generally sample high and medium ($w_{rise} = 0.03 - 0.003$ m s$^{-1}$) buoyancy particulates, which for non-biofouled polyethylene would

**Table 1.** Overview of the sources of field measurements of microplastic concentration profiles. The uncertainty in the mean MLD is the standard deviation.

| Source | Measurement Approach | Number of concentration profiles | Number of data points | Mean MLD [min max] (z) |
|---|---|---|---|---|
| Kooi et al. (2016b) | Neuston net | 46 | 506 | 15.4±3.6 [10.0, 26.2] |
| Pieper et al. (2019) | Niskin bottles | 12 | 152 | 17.1±5.5 [11.0, 28.0] |
| Kukulka et al. (2012) | Neuston net | 13 | 47 | 24.3±8.9 [11.0, 45.1] |
| Egger et al. (2020) | Neuston net | 16 | 20 | 55.8±19.2 [12.3, 72.8] |
| Amaral-Zettler (unpublished data) | Neuston net | 3 | 16 | 17.8±4.8 [14.0, 26.0] |
| **Total** | | 90 | 741 | 17.5±8.8 [10.0, 72.8] |

have a diameter greater than the mesh size (2.2 and 0.4 mm). In contrast, low buoyancy particulates ($w_{rise} = 0.0003$ m s$^{-1}$) are typically not sampled in neuston nets (Kooi et al., 2016b), likely in part due to smaller particulate sizes. Pieper et al. (2019) filtered samples collected via Niskin bottles with a $0.8\mu$m filter and thus was able to filter out smaller particulates with lower rise velocities.

195

All measured microplastic concentrations are normalized by total amount of plastic measured within a vertical profile. In order to compare the average normalized field concentration with the modelled profiles, we bin the normalized field concentrations into 0.5 m depth bins and calculate the standard deviation for each depth bin. Comparison of the modelled concentration profiles with the binned normalized field measurements is done via the root mean square error (RMSE):

$$200 \quad RMSE = \sqrt{\frac{1}{n}\sum_{i=0}^{n}(C_{f,i} - C_{m,i})} \tag{13}$$

where $C_{f,i}$ and $C_{m,i}$ are the binned normalized field measurement and modelled concentration within depth bin $i$. Model evaluation for the low buoyancy particles is not possible with the available field measurements as low buoyancy particles are typically too small to be sampled with neuston nets, and the Pieper et al. (2019) dataset alone is too small.

## 3 Results

205 Starting with all particles at $z = 0$ for $t = 0$, M-0 models with both KPP and SWB diffusion lead to stable vertical concentration profiles (Fig. 2), where the equilibrium concentration profile is already established within 1 - 2 hours (Fig. C1). For both diffusion profiles, there is progressively deeper mixing of particles with increasing wind speeds and decreasing buoyancy. While with both SWB and KPP diffusion low buoyancy particles always get mixed below the surface, for medium and high buoyancy particles there exist minimum wind speeds below which all particles remain at the surface. These limits are similar

210 for both diffusion types for medium buoyancy particles ($u_{10} \geq 2.40$ m s$^{-1}$), but high buoyancy particles only mix below the

surface with SWB diffusion if $u_{10} \geq 9.30$ m s$^{-1}$. However, once mixing below the ocean surface occurs, KPP diffusion always leads to deeper mixing of particles than SWB diffusion due to higher subsurface $K_z$ values.

The concentration profiles for medium and low buoyancy particles are largely unaffected by reducing $\Delta t$ below 30 seconds (Fig. F1). However, for high buoyancy particles with SWB diffusion the concentration profile more strongly depends on $\Delta t$ due to the applied boundary condition. For $\Delta t = 30$ s, the M-0 model shows all particles remain near the ocean surface, but shorter $\Delta t$ values indicate that deeper mixing of particles already occurs for $u_{10} = 6.65$ m s$^{-1}$. With KPP diffusion, all high buoyancy particles remain at the surface even with $\Delta t = 1$ second, as $K_z$ at $z = 0$ remains too low to overcome the high rise velocity.

Even though KPP diffusion with $\theta = 1.0$ and $z_0$ following (Zhao and Li, 2019) predicts deeper mixing of particles than with SWB diffusion ($\gamma = 1.0$), both approaches underpredict the mixing of particles relative to field observations. For KPP diffusion, this can be corrected by accounting for LC-driven mixing, which leads to deeper mixing of particles for both medium and low buoyancy particles (Figures 3 & D1). For medium buoyancy particles this generally leads to better model agreement with lower RMSE values between the modelled and averaged field data concentration profiles (Figure 5). However, for high buoyancy particles LC-driven circulation is not enough as particles remain at the ocean surface for all wind conditions even for $\theta = 5.0$ (Figure D2), as $K_z$ for $z \approx 0$ is too low to overcome the inherent particle buoyancy. Only when LC-driven is combined with higher near-surface $K_z$ values by setting $z_0 = 0.1 \times H_s$ do we see any below-surface mixing of high buoyancy particles when $\theta > 3.0$ and $u_{10} \geq 9.30$ m s$^{-1}$. Increased near-surface $K_z$ values have a lesser influence on the concentration profiles of medium and low density particles, as these particles were already being mixed below the surface even without larger $z_0$ values. For SWB diffusion we obtain deeper mixing of all particles by increasing $\gamma > 1.0$ (Figures 3, D1 & D2), which improves model performance relative to observations (Figure 5). While increasing $\gamma$ does not affect the peak magnitude of the near-surface $K_z$ values, it increases the depth until which $K_z$ is constant. This therefore results in stronger overall mixing (Figure 1), which in turn leads to the deeper mixing of the particles.

With both KPP and SWB diffusion, M-1 models show deeper mixing of particles as $\alpha \to 1$ (Fig. 6). Relative to the field measurements, M-1 models can at best slightly improve model performance over M-0 models (Fig. 7). However, improved model performance is not shown across all particle sizes and wind conditions, and there is not a consistent $\alpha$ value leading to the smallest RMSE values.

## 4 Discussion

The parametrizations presented in this study are intended for use in 3D Lagrangian experiments using OGCM data, and therefore should yield numerically stable results for the relatively large integration timesteps used in large-scale Lagrangian vertical transport modelling (Lobelle et al., 2021). While there are more stable schemes available than the EM scheme used in this study (Gräwe et al., 2012), the EM scheme is computationally the cheapest and yields concentration profiles that match reasonably

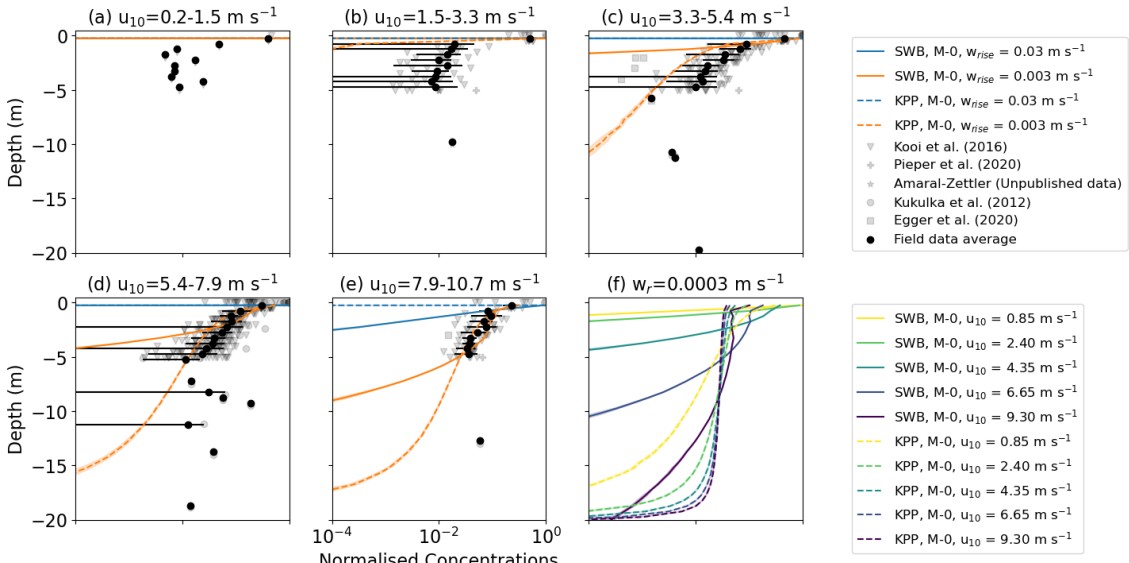

**Figure 2.** Vertical concentrations of buoyant particles for KPP and SWB diffusion using M-0 models. Subfigures (a) - (e) show the vertical concentration profiles for high and medium buoyancy particles with increasing wind speeds. The KPP profiles are calculated for $\theta = 1.0$ and $z_0$ according to Equation 12. The grey markers indicate field measurements, with darker shades indicating more measurements, while the binned field measurement average and standard deviation are shown by the black markers. Subfigure (f) shows the vertical concentration profiles for low buoyancy particles under increasing wind conditions. Shading around the profiles indicates the profile's standard deviation at each depth level.

well with observations. Both M-0 and M-1 models show largely convergent concentration profiles for $\Delta t = 30$ seconds, which
would make both approaches feasible with regards to computational cost. However, we would currently recommend using a M-0 model. M-1 models have the additional tuning parameter $\alpha$ representing the autocorrelation of turbulent velocity fluctuations, which is poorly constrained in the literature. Using spatially invariant $\alpha$ values at best slightly improved model performance in comparison with M-0 models, and constraining $\alpha$ is not possible from these results. M-1 models may improve modelling of vertical diffusive transport, but more work is required to further constrain the value and vertical profile of $\alpha$. Finally, numer-
ous formulations of the M-1 drift term have been proposed (Mofakham and Ahmadi, 2020; Brickman and Smith, 2002, e.g.) which can lead to large differences in the modelled profiles. In this study we used the non-normalized Langevin equation from Mofakham and Ahmadi (2020), but other formulations could be explored in future work.

While the concentration profiles of medium and low buoyancy particles are unaffected by decreasing the integration timestep
$\Delta t < 30$ seconds, using higher $\Delta t$ values underestimates the depth to which high buoyancy particles are mixed when using SWB diffusion. This is because for high $\Delta t$ values, the upward non-stochastic component of equation 6, which scales with $\Delta t$, dominates the stochastic component, which scales with $\sqrt{\Delta t}$. With KPP diffusion the vertical profile for high buoyancy

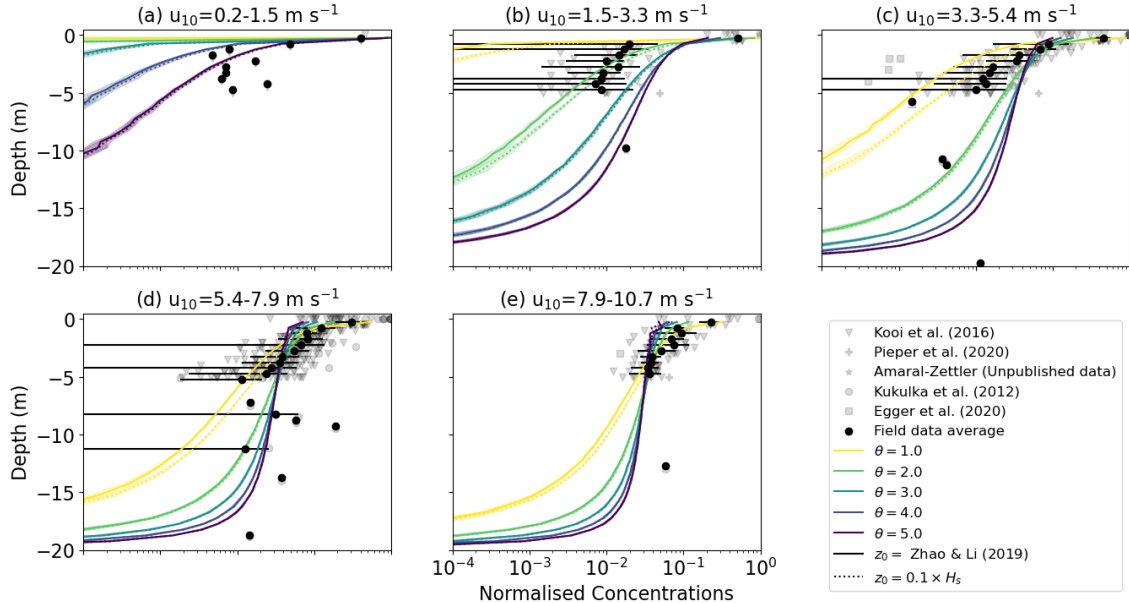

**Figure 3.** Vertical concentrations of buoyant particles for KPP diffusion using M-0 models for $w_r = 0.003$ m s$^{-1}$. The KPP profiles are calculated for $\theta = [1.0, 2.0, 3.0, 4.0, 5.0]$ and with either $z_0 = 0.1 \times H_s$ or according to Equation 12. The grey markers indicate field measurements, with darker shades indicating more measurements, while the binned field measurement average and standard deviation are shown by the black markers. Shading around the profiles indicates the profile's standard deviation at each depth level.

particles appears unaffected by $\Delta t$, but this is because the near-surface $K_z$ values are significantly lower than with SWB diffusion. One possibility to correct for this is to apply a different BC, such as a reflective BC. While the concentration profiles
for medium and low buoyancy particles are not strongly affected by such a reflective BC (Fig. G1), the reflective BC does show deeper particle mixing with SWB diffusion. However, for $\Delta t = 30$ seconds the depth of mixing is now overestimated compared to smaller $\Delta t$ values (Fig. G2), as with $\Delta t = 30$ seconds and $w_r = 0.03$ m s$^{-1}$ the particle would be reflected up to 0.9 m below the ocean surface solely due to the model numerics. In addition, earlier studies have shown that reflecting BC can cause spurious increases in particle concentration near the boundary (Ross and Sharples, 2004; Nordam et al., 2019). There-
fore, changing the BC to a reflective BC would not improve the concentration profiles of high buoyancy particles. Depending on the model application and setup, the error in the concentration profile depth ($\mathcal{O}(1)$ m for high buoyancy particles) might be acceptable. Otherwise, the error can be reduced by using a smaller integration timestep where that is computationally feasible.

Considering the KPP and SWB diffusion profiles, the results in this study are inconclusive with regards to which approach
performs better relative to field observations. For high buoyancy particles, SWB diffusion leads to slightly deeper particle mixing, while only if the KPP diffusion profile accounts for LC-driven turbulence and has higher near-surface $K_z$ values can it similarly show below-surface mixing of high buoyancy particles for $u_{10} \geq 9.30$ m s$^{-1}$. With medium and low buoyancy

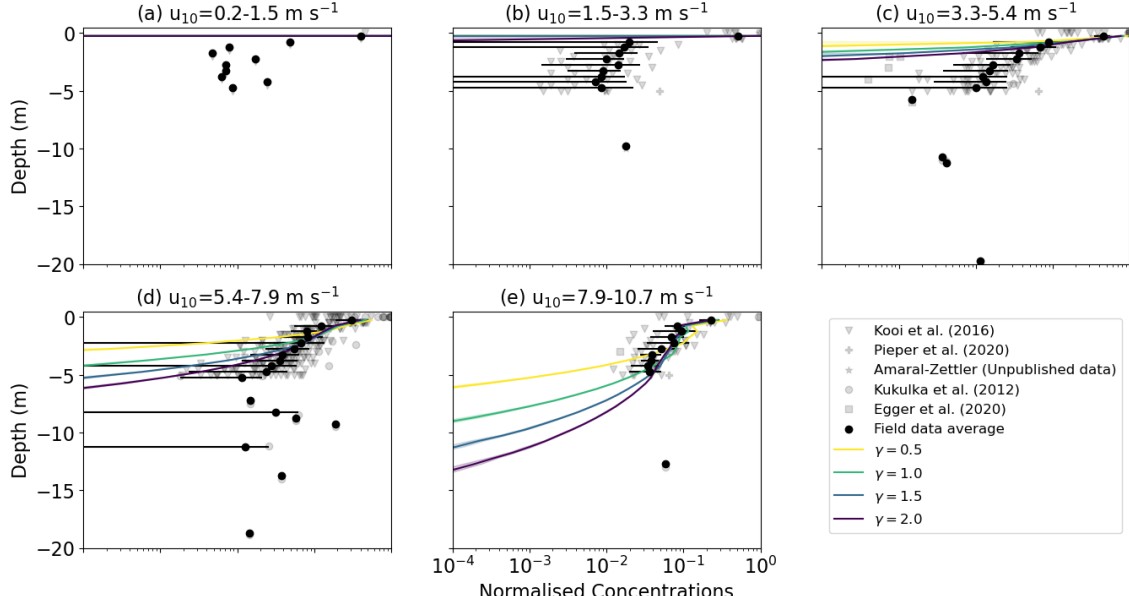

**Figure 4.** Vertical concentrations of buoyant particles for SWB diffusion under varying wind conditions with $w_r = 0.003$ m s$^{-1}$. The SWB diffusion profile is calculated with $\gamma \in [0.5, 1.0, 1.5, 2.0]$. The grey markers indicate field measurements, with darker shades indicating more measurements, while the binned field measurement average and standard deviation are shown by the black markers. Shading around the profiles indicates the profile's standard deviation at each depth level.

particles the KPP profile leads to much deeper mixing compared with SWB diffusion where $\gamma = 1.0$ Poulain (2020), especially when accounting for LC-driven turbulence, and this appears to agree better with field observations. However, for SWB diffu-
sion the value of $\gamma$ is uncertain, as Poulain (2020) and Kukulka et al. (2012) respectively define $\gamma = 1.0$ and $\gamma \approx 1.5$. Higher $\gamma$ values leads to approximately equal model performance relative to field observations as with KPP diffusion. However, the model evaluation is largely based on field measurements collected in the top 5 meters of the water column, and it is below this depth that we see greater differences in the KPP and SWB vertical concentration profiles. In addition, the currently available data collected with Neuston nets does not allow for model evaluation for the low-buoyancy particles in either scenario. As
such, more field measurements (including smaller-sized particles) would be necessary to fully evaluate model performance for all particles sizes with the two diffusion profiles.

With regards to necessary data to calculate the diffusion profiles, the SWB approach has the benefit that it only requires surface wind stress data, while KPP diffusion additionally requires MLD data. In addition, while our results indicate that ac-
counting for LC-driven turbulent mixing improves KPP diffusion model performance, determining which $\theta$ value to use is not trivial. McWilliams and Sullivan (2000) demonstrated that $\theta$ is inversely proportional to the Langmuir number $La$, which is defined as $La = \sqrt{u_{*w}/U_S}$ with $U_S$ as the surface Stokes drift. The Langmuir number can conceivably be calculated using

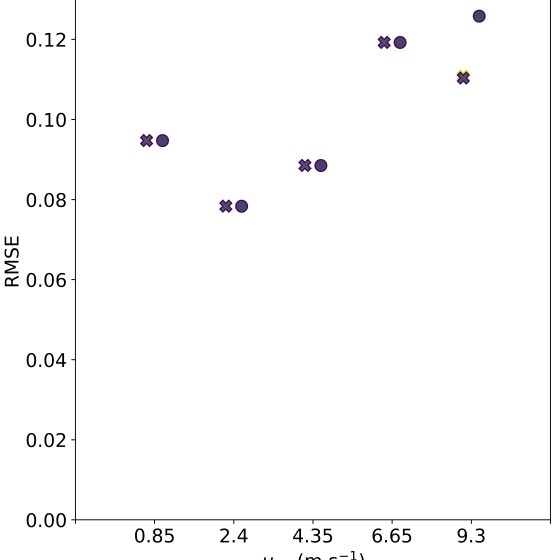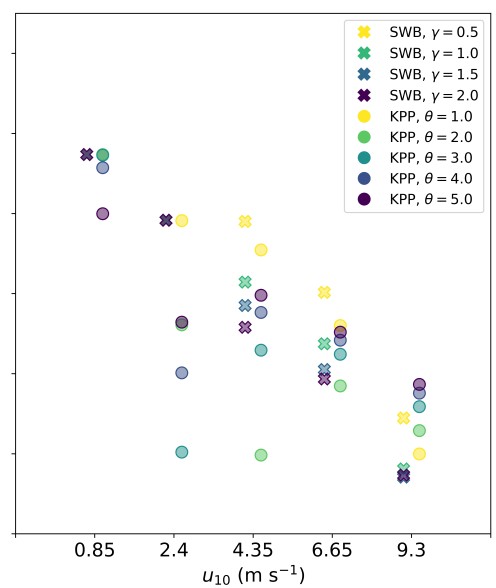

**Figure 5.** RMSE between field measurements and modelled concentration profiles for M-0 models with KPP and SWB diffusion under different wind conditions. All KPP diffusion simulations were with $z_0$ according to Equation 12.

OGCM data, but the details of such an implementation will be left for future work with 3D Lagrangian models. However, KPP diffusion does have the advantage that it has been widely used and validated in various model setups (Boufadel et al., 2020; McWilliams and Sullivan, 2000; Large et al., 1994), while such extensive validation has not yet occured for SWB diffusion. Finally, the influence of wind forcing on turbulence is generally assumed to be limited to the surface mixed layer (Chamecki et al., 2019), while with the SWB profile wind-generated turbulence can extend far below the MLD (Figures 1 & E1), possibly overestimating turbulent mixing at such depths. KPP theory does limit wind-driven turbulent mixing to the surface mixed layer, while either a constant $K_z$ value or other $K_z$ profiles could be used for sub-MLD mixing, such as the $K_z$ estimates for internal tide mixing as proposed by de Lavergne et al. (2020).

Ideally, KPP theory would be expanded to account for surface wave breaking, which could lead to higher near-surface $K_z$ values as seen with MLD diffusion. While such a theoretical approach is beyond the scope of this paper, we show that artificially elevating near-surface $K_z$ values by increasing the surface roughness $z_0$ has a smaller influence on the overall concentration profile than LC-driven mixing, as similarly shown by (Brunner et al., 2015). Therefore, although we recommend future work incorporating surface wave breaking into KPP theory, our current KPP diffusion approach representing LC-driving mixing through $\theta$ does already seem to capture the majority of turbulent mixing dynamics.

In all cases, the vertical concentration profiles stabilized to vertical equilibrium profiles, similar to what has been shown for buoyant particles in LES model studies (Liang et al., 2012; Yang et al., 2014; Brunner et al., 2015; Taylor, 2018). The modelled

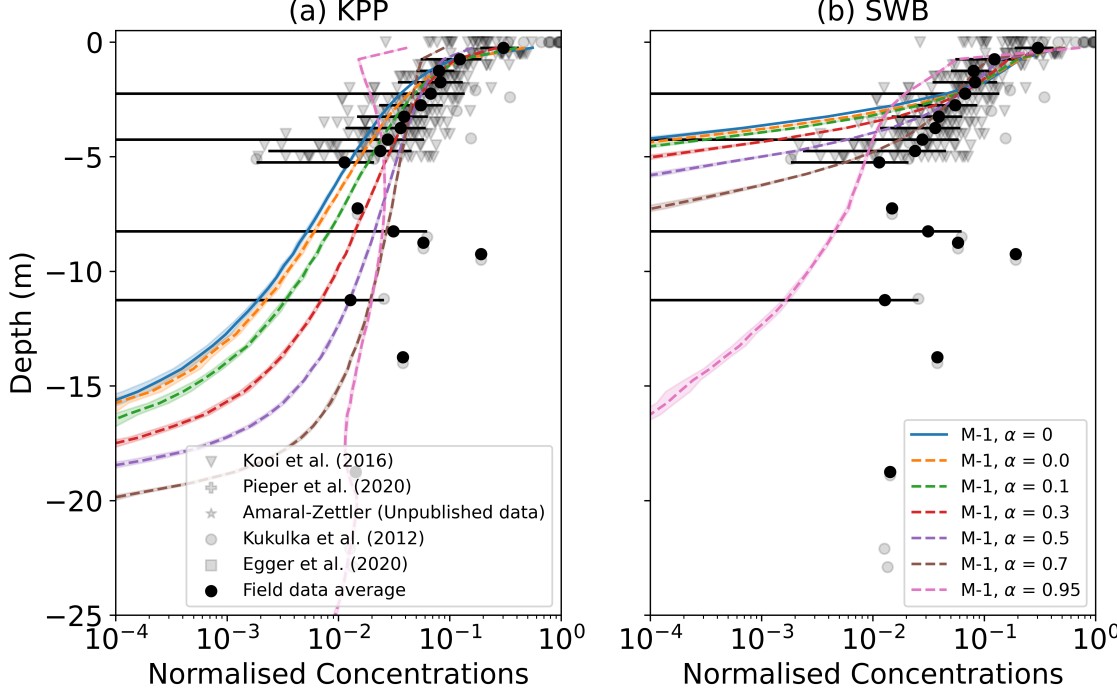

**Figure 6.** Vertical concentrations of buoyant particles for (a) KPP and (b) SWB diffusion using M-0 and M-1 models with varying values for $\alpha$. The grey markers indicate field measurements, with darker shades indicating more measurements, while the binned field measurement average and standard deviation are shown by the black markers. Shading around the profiles indicates the profile's standard deviation at each depth level. The KPP profiles are for $\theta = 1.0$ and $z_0$ according to 12. All profiles are for $u_{10} = 6.65$ m s$^{-1}$ and medium buoyancy particles ($w_{rise} = 0.003$ m s$^{-1}$).

concentration profiles generally resembled the profiles from field measurements of microplastic concentrations under different wind conditions (Kooi et al., 2016b; Kukulka et al., 2012), but the averaged concentration profiles of the field measurements are quite noisy. Partly, this could be due to inhomogeneity in the particle buoyancy, as the collected microplastic particulates have varying sizes and rise velocities (Kooi et al., 2016b; Egger et al., 2020). Additionally, we sorted the field measurements
based on wind conditions, but other underlying oceanographic conditions such as the MLD can still vary significantly even with similar wind speeds. Unfortunately, we lack additional data of the oceanographic conditions at the of sampling, which currently prohibits more high-level comparisons of the field and model concentration profiles. Compared with the field data, the variance in the modelled concentration profiles is significantly smaller. This is in part also due to assuming constant environmental conditions over 12 hours for the model simulations, while wind and other oceanographic conditions can change on
much shorter timescales over the ocean surface. To further improve vertical transport model verification, more measurements would be required, covering a wider range of oceanographic conditions (such as for wind conditions higher than $u_{10} = 10.7$ m s$^{-1}$) and with a high spatial sampling resolution also for depths $z < -5$m. Ideally these measurements would also sample

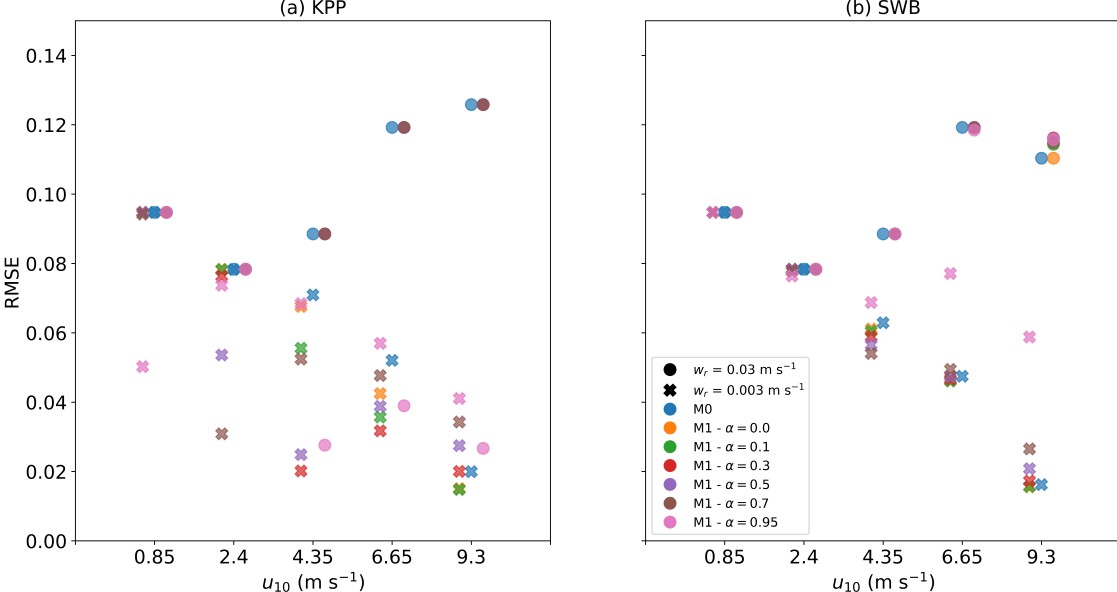

**Figure 7.** RMSE between field measurements and modelled concentration profiles for M-0 and M-1 models with (a) KPP and (b) SWB diffusion under different wind conditions and with varying values of $\alpha$. All KPP diffusion simulations were with $\theta = 1.0$ and $z_0$ according to 12.

small, neutrally buoyant particulates, but we acknowledge this is difficult with the sampling techniques commonly used today. At the same time, we would encourage conducting more ocean field measurements of near-surface vertical eddy diffusion co-

efficient and/or eddy viscosity profiles, as this will allow further validation of the $K_z$ profiles predicted by the KPP and SWB theory with actual ocean near-surface mixing measurements.

The parameterizations have been validated for high/medium rise velocities, and at least for KPP diffusion with $\theta > 1.0$ the concentration profiles resemble those calculated from field observations. This provides confidence in the turbulence estimates

from the KPP approach, and as these are independent of the type of particle that might be present, this would suggest the KPP approach can also be applied to to neutral or negatively buoyant particles. However, as model verification was only possible for microplastic particulates with rise velocities approximately between 0.03 - 0.003 m s$^{-1}$, we would advise additional model verification for other particle types where the necessary field data is available. In the case of SWB diffusion, turbulent mixing seems underestimated when further from the ocean surface when $\gamma = 1.0$, but increasing to $\gamma = 1.5 - 2.0$ does correct for this.

However, as SWB diffusion has not yet been as extensively tested and verified as KPP diffusion, we advice more caution and additional validation with field observations before applying this diffusion approach to other particle types.

## 5 Conclusions

We have developed a number of 1D surface-mixing parametrizations designed to be readily applied in large-scale oceanic Lagrangian model experiments using OGCM data. Where possible, we would recommend using the turbulence fields from the OGCM to assure turbulent transport of the particles is consistent with that of other model tracers. However, if the turbulence fields are unavailable then particularly parametrizations with KPP diffusion with LC-driven mixing are shown to produce modelled vertical concentration profiles that match relatively well with field observations of microplastics. The parametrizations generally perform well for timesteps of $\Delta t = 30$ seconds, but for high buoyancy particles users need to take care to use sufficiently short timesteps, especially with SWB diffusion. Verification was only possible for positively buoyant particles larger than $0.33$ mm (which generally have rise velocities $\leq 0.003$ m s$^{-1}$), but the parametrizations should also be applicable to other particle types. The parametrizations can therefore be applied to investigate the influence of turbulent mixing on the vertical transport of (microplastic) particles within a 3D model setup, and ultimately gain a more complete understanding of the fate of such particles in the ocean.

## 6 Code and data availability

The code for the 1D model, the subsequent analysis and all figures is available at zenodo (Onink, 2021). The field data for Kooi et al. (2016b) is available at figshare (Kooi et al., 2016a). For the field data from Kukulka et al. (2012), Pieper et al. (2019), Egger et al. (2020) and Amaral-Zettler (unpublished data), please contact the corresponding authors of the respective studies.

**Table A1.** Ratios $w_r/w'$ between the rise velocity $w_r$ and the peak stochastic velocity perturbation $w'$ for KPP and SWB diffusion. The peak $w'$ is the maximum value of Equation 3. The peak $w'$ values for KPP diffusion are calculated for $\theta \in [1.0, 3.0, 5.0]$ and for $z_0$ following Equation 12. The peak $w'$ values for SWB diffusion are independent of $\gamma$.

| Wind Speed (m s$^{-1}$) | Diffusion Type | $w_r = 0.03$ m s$^{-1}$ | $w_r = 0.003$ m s$^{-1}$ | $w_r = 0.0003$ m s$^{-1}$ |
|---|---|---|---|---|
| 0.85 | KPP, $\theta = 1.0$ | 1.818 | 0.182 | 0.018 |
| | KPP, $\theta = 3.0$ | 1.055 | 0.106 | 0.011 |
| | KPP, $\theta = 5.0$ | 0.818 | 0.082 | 0.008 |
| | SWB | 10.512 | 1.051 | 0.105 |
| 2.40 | KPP, $\theta = 1.0$ | 1.087 | 0.109 | 0.011 |
| | KPP, $\theta = 3.0$ | 0.628 | 0.063 | 0.006 |
| | KPP, $\theta = 5.0$ | 0.486 | 0.049 | 0.005 |
| | SWB | 4.077 | 0.408 | 0.041 |
| 4.35 | KPP, $\theta = 1.0$ | 0.808 | 0.081 | 0.008 |
| | KPP, $\theta = 3.0$ | 0.465 | 0.047 | 0.005 |
| | KPP, $\theta = 5.0$ | 0.359 | 0.036 | 0.004 |
| | SWB | 1.753 | 0.175 | 0.018 |
| 6.65 | KPP, $\theta = 1.0$ | 0.654 | 0.065 | 0.007 |
| | KPP, $\theta = 3.0$ | 0.373 | 0.037 | 0.004 |
| | KPP, $\theta = 5.0$ | 0.288 | 0.029 | 0.003 |
| | SWB | 0.935 | 0.094 | 0.009 |
| 9.30 | KPP, $\theta = 1.0$ | 0.553 | 0.055 | 0.006 |
| | KPP, $\theta = 3.0$ | 0.313 | 0.031 | 0.003 |
| | KPP, $\theta = 5.0$ | 0.241 | 0.024 | 0.002 |
| | SWB | 0.566 | 0.057 | 0.006 |

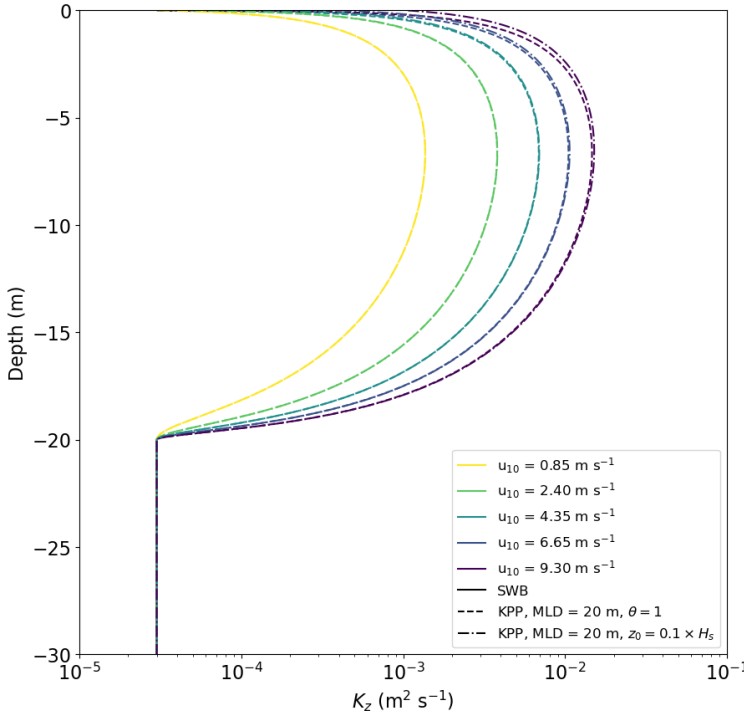

**Figure B1.** Vertical diffusion coefficient profiles for KPP diffusion under varying wind conditions with $\theta = 1.0$. The KPP diffusion profile is calculated either with $z_0$ according to Equation 12 or $z_0 = 0.1 \times H_s$.

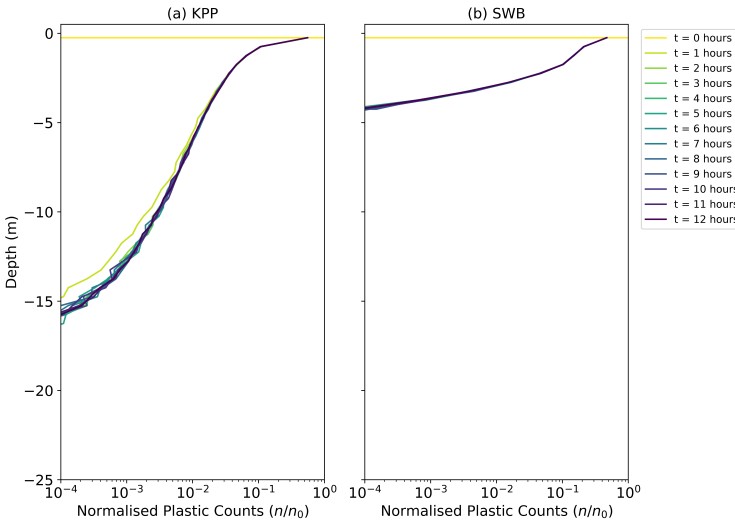

**Figure C1.** Vertical concentrations of buoyant particles for KPP diffusion at times $t = 0 - 12$ hours. The KPP diffusion profile is calculated with $\theta = 1.0$, $u_{10} = 6.65$ m s$^{-1}$, and $z_0$ according to Equation 12.

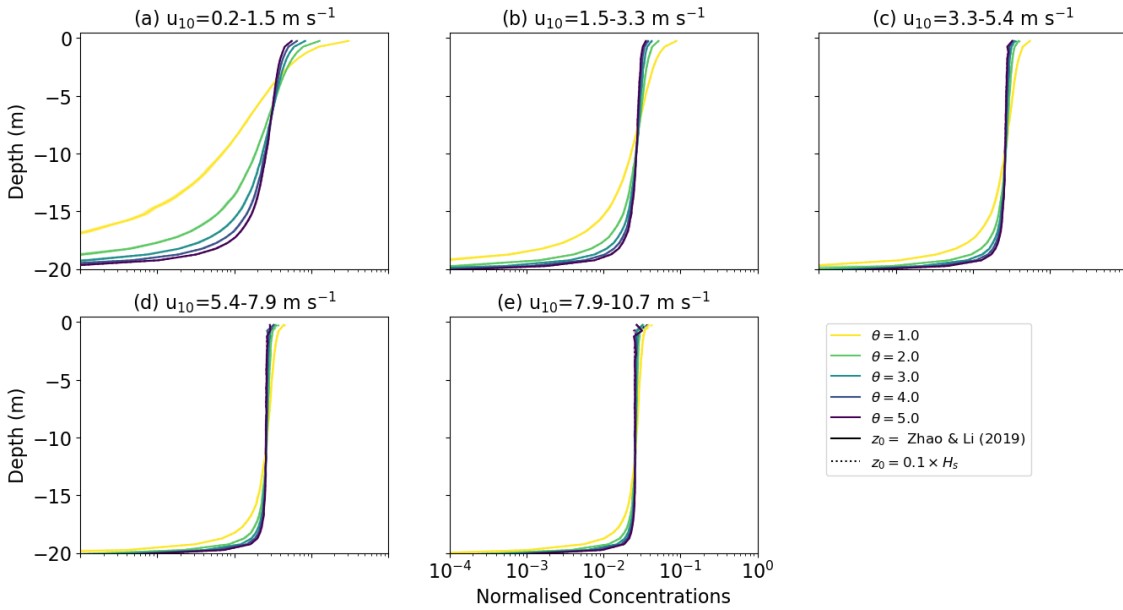

**Figure D1.** Vertical concentrations of buoyant particles for KPP diffusion under varying wind conditions with $w_r = 0.0003$ m s$^{-1}$. The KPP diffusion profile is calculated either with $z_0$ according to Equation 12 or $z_0 = 0.1 \times H_s$, and for $\theta \in [1.0, 2.0, 3.0, 4.0, 5.0]$. The grey markers indicate field measurements, with darker shades indicating more measurements, while the binned field measurement average and standard deviation are shown by the black markers. Shading around the profiles indicates the profile's standard deviation at each depth level.

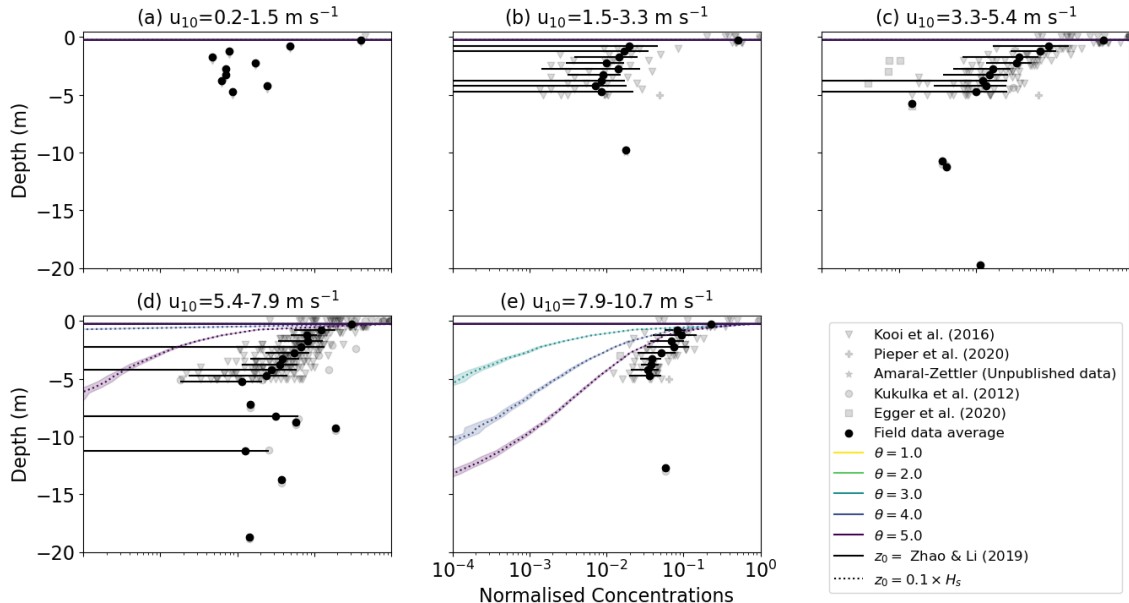

**Figure D2.** Vertical concentrations of buoyant particles for KPP diffusion under varying wind conditions with $w_r = 0.03$ m s$^{-1}$. The KPP diffusion profile is calculated either with $z_0$ according to Equation 12 or $z_0 = 0.1 \times H_s$, and for $\theta \in [1.0, 2.0, 3.0, 4.0, 5.0]$. The grey markers indicate field measurements, with darker shades indicating more measurements, while the binned field measurement average and standard deviation are shown by the black markers. Shading around the profiles indicates the profile's standard deviation at each depth level.

**Appendix A: $w_r/w'$ ratios for various turbulence scenarios**

**Appendix B: Influence of $z_0$ on diffusion profiles**

**Appendix C: Time evolution of concentration profiles**

**Appendix D: Influence of $\theta$ for KPP diffusion**

**Appendix E: Influence of $\gamma$ for SWB diffusion**

**Appendix F: Influence of $\Delta t$**

**Appendix G: Influence of boundary conditions**

*Author contributions.* Development of the parametrizations and the analysis was done by VO, with CL helping with improving the code performance. The manuscript was written by VO, with extensive input from CL and EvS. Everyone contributed to the study design and discussion of the analysis.

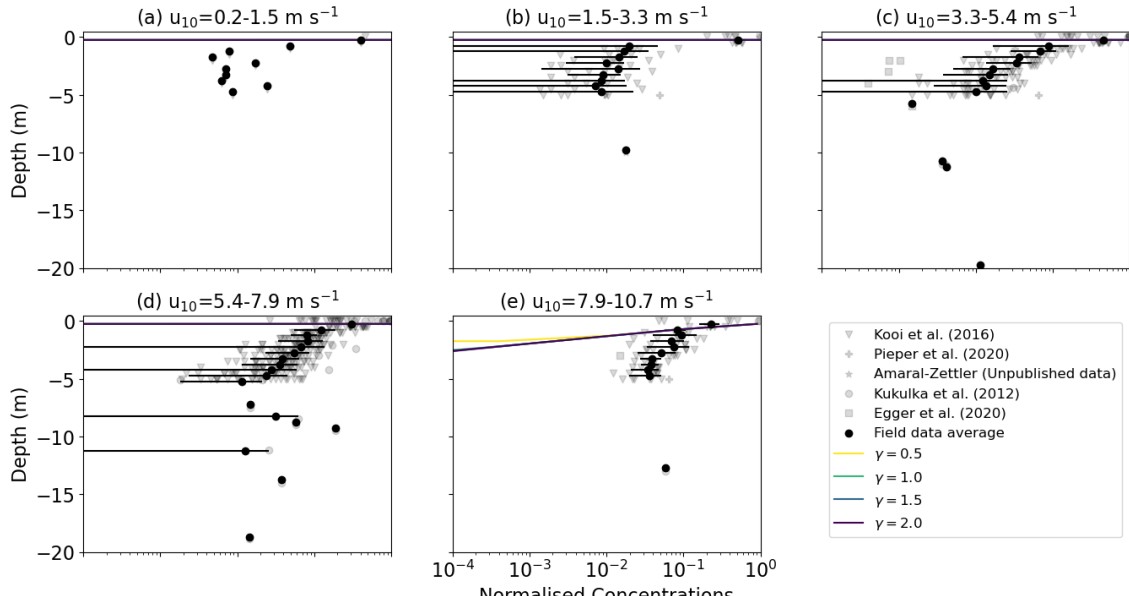

**Figure E1.** Vertical concentrations of buoyant particles for SWB diffusion under varying wind conditions with $w_r = 0.0003$ m s$^{-1}$. The SWB diffusion profile is calculated with $\gamma \in [0.5, 1.0, 1.5, 2.0]$. The grey markers indicate field measurements, with darker shades indicating more measurements, while the binned field measurement average and standard deviation are shown by the black markers. Shading around the profiles indicates the profile's standard deviation at each depth level.

*Competing interests.* The authors declare no competing interests.

*Acknowledgements.* VO and CL acknowledge support from the Swiss National Science Foundation (project PZ00P2_174124 Global interactions between microplastics and marine ecosystems). EvS was supported by the European Research Council (ERC) under the European Unions Horizon 2020 research and innovation programme (grant agreement No 715386). Calculations were performed on UBELIX (http://www.id.unibe.ch/hpc), the HPC cluster at the University of Bern. We would like to thank Dr. Tobias Kukulka, Dr. Catharina Pieper, Dr. Matthias Egger, Dr. Linda Amaral-Zettler and Dr. Erik Zettler for providing field measurements, Dr. Marie Poulain-Zarcos for providing data on vertical mixing, and Dr. Thomas Stocker and Daan Reijnders for fruitful discussions regarding the Markov-1 numerical schemes.


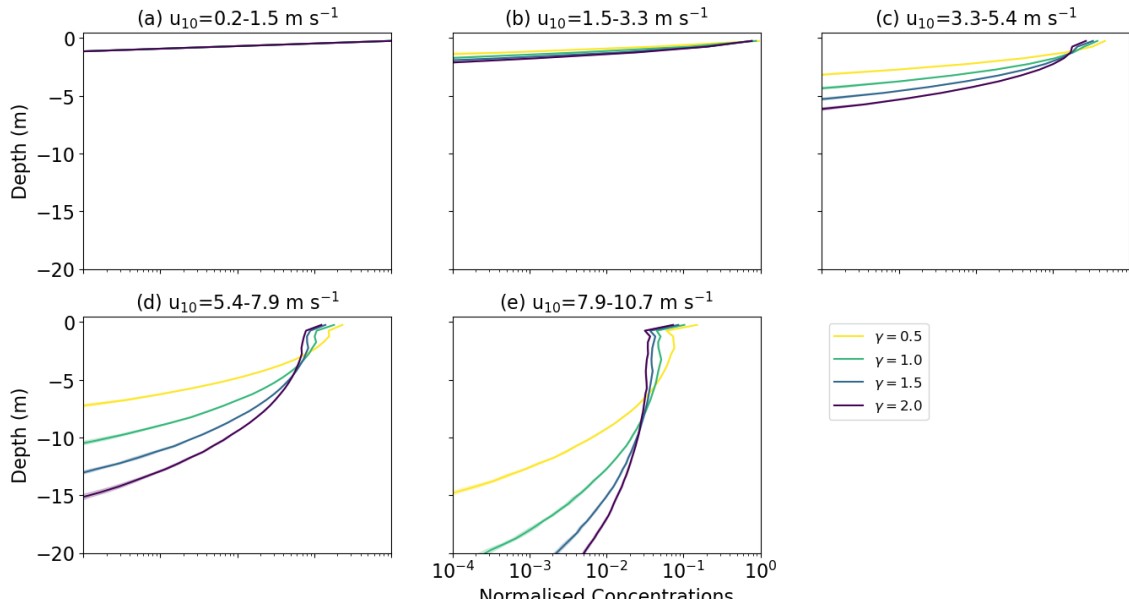

**Figure E2.** Vertical concentrations of buoyant particles for KPP diffusion under varying wind conditions with $w_r = 0.03$ m s$^{-1}$. The SWB diffusion profile is calculated with $\gamma \in [0.5, 1.0, 1.5, 2.0]$. The grey markers indicate field measurements, with darker shades indicating more measurements, while the binned field measurement average and standard deviation are shown by the black markers. Shading around the profiles indicates the profile's standard deviation at each depth level.

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

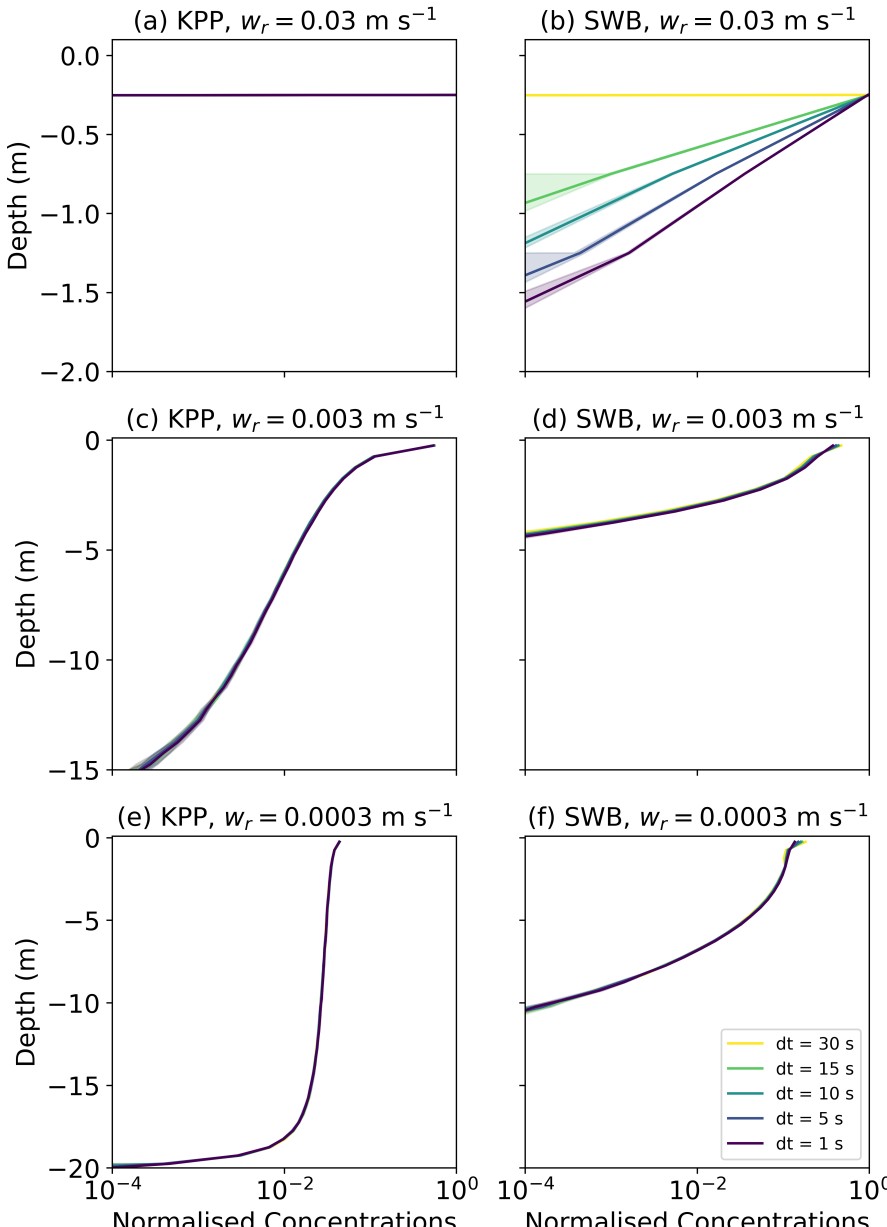

**Figure F1.** Vertical concentrations of buoyant particles for (a, c, e) KPP and (b, d, f) SWB diffusion using M-0 models with varying values for $w_{rise}$ and $\Delta t \in [30, 15, 10, 5, 1]$ second(s). All profiles are for $u_{10} = 6.65$ m s$^{-1}$. Shading around the profiles indicates the profile's standard deviation at each depth level. The KPP profiles are computed with $\theta = 1.0$ and $z_0$ according to Equation 12, while the SWB profile is computed with $\gamma = 1.0$.

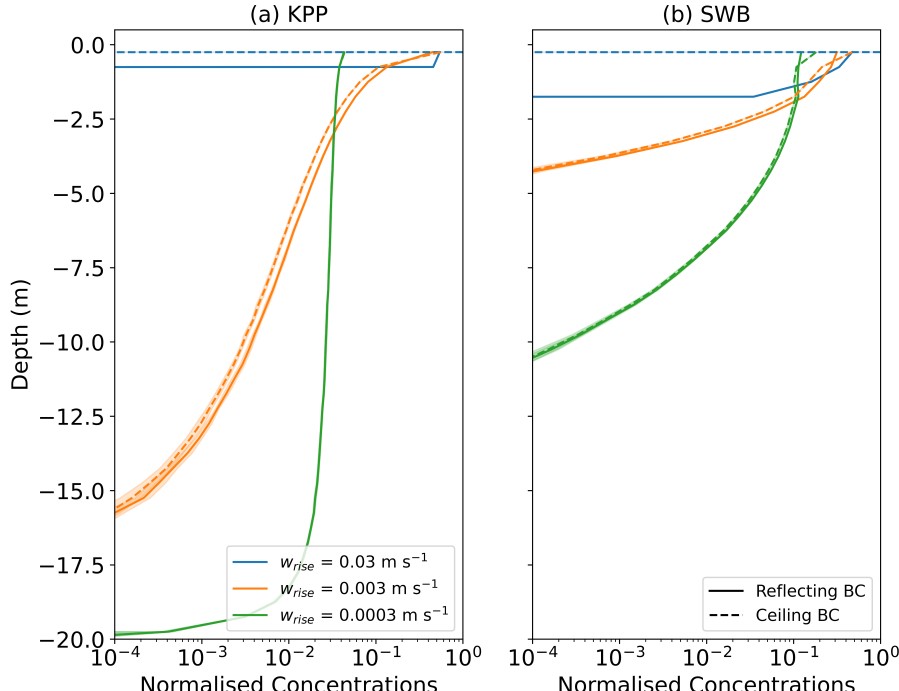

**Figure G1.** Vertical concentrations of buoyant particles for (a) KPP and (b) SWB diffusion using M-0 models for reflective and ceiling BC's. Shading around the profiles indicates the profile's standard deviation at each depth level. All profiles are for $u_{10} = 6.65$ m s$^{-1}$. The KPP profiles are computed with $\theta = 1.0$ and $z_0$ according to Equation 12, while the SWB profile is computed with $\gamma = 1.0$.

Craig, P. D. and Banner, M. L.: Modeling wave-enhanced turbulence in the ocean surface layer, Journal of Physical Oceanography, 24, 2546–2559, 1994.

de Boyer Montégut, C., Madec, G., Fischer, A. S., Lazar, A., and Iudicone, D.: Mixed layer depth over the global ocean: An examination of profile data and a profile-based climatology, Journal of Geophysical Research: Oceans, 109, 2004.

de Lavergne, C., Vic, C., Madec, G., Roquet, F., Waterhouse, A. F., Whalen, C., Cuypers, Y., Bouruet-Aubertot, P., Ferron, B., and Hibiya, T.: A parameterization of local and remote tidal mixing, Journal of Advances in Modeling Earth Systems, 12, e2020MS002 065, 2020.

Delandmeter, P. and Sebille, E. v.: The Parcels v2. 0 Lagrangian framework: new field interpolation schemes, Geoscientific Model Development, 12, 3571–3584, 2019.

Denman, K. and Gargett, A.: Time and space scales of vertical mixing and advection of phytoplankton in the upper ocean, Limnology and

oceanography, 28, 801–815, 1983.

Egger, M., Sulu-Gambari, F., and Lebreton, L.: First evidence of plastic fallout from the North Pacific Garbage Patch, Scientific reports, 10, 1–10, 2020.

Enders, K., Lenz, R., Stedmon, C. A., and Nielsen, T. G.: Abundance, size and polymer composition of marine microplastics$\geq 10$ $\mu$m in the Atlantic Ocean and their modelled vertical distribution, Marine pollution bulletin, 100, 70–81, 2015.

Fernando, H. J.: Turbulent mixing in stratified fluids, Annual review of fluid mechanics, 23, 455–493, 1991.

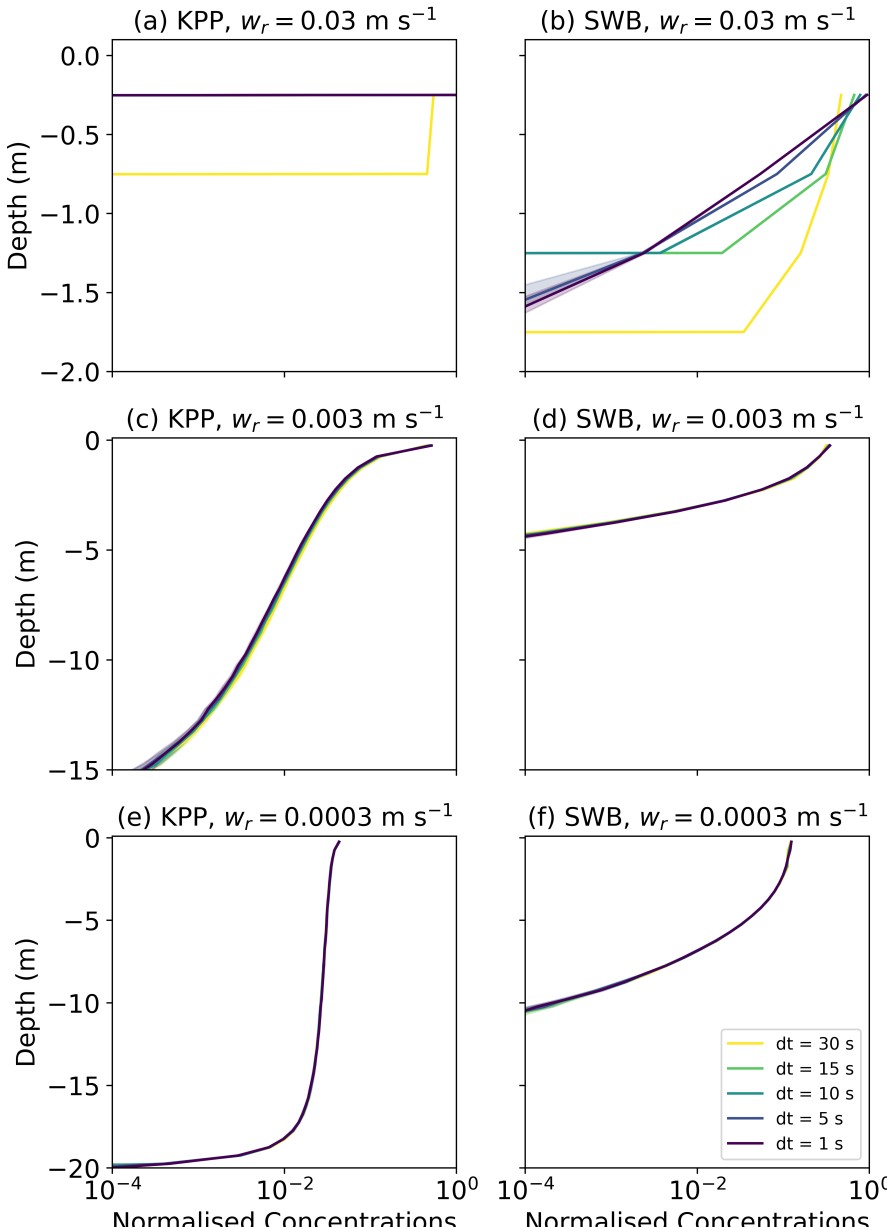

**Figure G2.** Vertical concentrations of buoyant particles for (a, c, e) KPP and (b, d, f) SWB diffusion using M-0 models with varying values for $w_{rise}$ and $\Delta t \in [30, 15, 10, 5, 1]$ second(s) with a reflective BC. All profiles are for $u_{10} = 6.65$ m s$^{-1}$. Shading around the profiles indicates the profile's standard deviation at each depth level. The KPP profiles are computed with $\theta = 1.0$ and $z_0$ according to Equation 12, while the SWB profile is computed with $\gamma = 1.0$.

Fischer, R., Lobelle, D., Kooi, M., Koelmans, A., Onink, V., Laufkötter, C., Amaral-Zettler, L., Yool, A., and van Sebille, E.: Modeling submerged biofouled microplastics and their vertical trajectories, Biogeosciences Discussions, pp. 1–29, 2021.

Gaspar, P., Grégoris, Y., and Lefevre, J.-M.: A simple eddy kinetic energy model for simulations of the oceanic vertical mixing: Tests at station Papa and Long-Term Upper Ocean Study site, Journal of Geophysical Research: Oceans, 95, 16 179–16 193, 1990.

Gräwe, U., Deleersnijder, E., Shah, S. H. A. M., and Heemink, A. W.: Why the Euler scheme in particle tracking is not enough: the shallow-sea pycnocline test case, Ocean Dynamics, 62, 501–514, 2012.

Hopfinger, E. and Toly, J.-A.: Spatially decaying turbulence and its relation to mixing across density interfaces, Journal of fluid mechanics, 78, 155–175, 1976.

Kaiser, D., Kowalski, N., and Waniek, J. J.: Effects of biofouling on the sinking behavior of microplastics, Environmental Research Letters, 405 12, 124 003, 2017.

Kooi, M., Reisser, J., Slat, B., Ferrari, F., Schmid, M., Cunsolo, S., Brambini, R., Noble, K., Sirks, L.-A., Linders, T. E., Schoeneich-Argent, R. I., and Koelmans, A. A.: Data from 'The effect of particle properties on the depth profile of buoyant plastics in the ocean', https://doi.org/10.6084/m9.figshare.3427862.v1, 2016a.

Kooi, M., Reisser, J., Slat, B., Ferrari, F. F., Schmid, M. S., Cunsolo, S., Brambini, R., Noble, K., Sirks, L.-A., Linders, T. E., et al.: The 410 effect of particle properties on the depth profile of buoyant plastics in the ocean, Scientific reports, 6, 1–10, 2016b.

Kukulka, T. and Brunner, K.: Passive buoyant tracers in the ocean surface boundary layer: 1. Influence of equilibrium wind-waves on vertical distributions, Journal of Geophysical Research: Oceans, 120, 3837–3858, 2015.

Kukulka, T. and Veron, F.: Lagrangian investigation of wave-driven turbulence in the ocean surface boundary layer, Journal of Physical Oceanography, 49, 409–429, 2019.

Kukulka, T., Proskurowski, G., Morét-Ferguson, S., Meyer, D., and Law, K.: The effect of wind mixing on the vertical distribution of buoyant plastic debris, Geophysical Research Letters, 39, 2012.

Landahl, M. T. and Christensen, E. M.: Turbulence and random processes in fluid mechanics, Cambridge University Press, 1998.

Large, W. and Pond, S.: Open ocean momentum flux measurements in moderate to strong winds, Journal of physical oceanography, 11, 324–336, 1981.

Large, W. G., McWilliams, J. C., and Doney, S. C.: Oceanic vertical mixing: A review and a model with a nonlocal boundary layer parameterization, Reviews of geophysics, 32, 363–403, 1994.

Law, K. L., Moré´t-Ferguson, S. E., Goodwin, D. S., Zettler, E. R., DeForce, E., Kukulka, T., and Proskurowski, G.: Distribution of surface plastic debris in the eastern Pacific Ocean from an 11-year data set, Environmental science & technology, 48, 4732–4738, 2014.

Lebreton, L., Slat, B., Ferrari, F., Sainte-Rose, B., Aitken, J., Marthouse, R., Hajbane, S., Cunsolo, S., Schwarz, A., Levivier, A., et al.: 425 Evidence that the Great Pacific Garbage Patch is rapidly accumulating plastic, Scientific reports, 8, 1–15, 2018.

Liang, J.-H., McWilliams, J. C., Sullivan, P. P., and Baschek, B.: Large eddy simulation of the bubbly ocean: New insights on subsurface bubble distribution and bubble-mediated gas transfer, Journal of Geophysical Research: Oceans, 117, 2012.

Liubartseva, S., Coppini, G., Lecci, R., and Clementi, E.: Tracking plastics in the Mediterranean: 2D Lagrangian model, Marine pollution bulletin, 129, 151–162, 2018.

Lobelle, D., Kooi, M., Koelmans, A. A., Laufkötter, C., Jongedijk, C. E., Kehl, C., and van Sebille, E.: Global modeled sinking characteristics of biofouled microplastic, Journal of Geophysical Research: Oceans, 126, e2020JC017 098, 2021.

Maruyama, G.: Continuous Markov processes and stochastic equations, Rendiconti del Circolo Matematico di Palermo, 4, 48, 1955.

McWilliams, J. C. and Sullivan, P. P.: Vertical mixing by Langmuir circulations, Spill Science & Technology Bulletin, 6, 225–237, 2000.

Mofakham, A. A. and Ahmadi, G.: On random walk models for simulation of particle-laden turbulent flows, International Journal of Multi-phase Flow, 122, 103 157, 2020.

Nordam, T., Kristiansen, R., Nepstad, R., and Röhrs, J.: Numerical analysis of boundary conditions in a Lagrangian particle model for vertical mixing, transport and surfacing of buoyant particles in the water column, Ocean Modelling, 136, 107–119, 2019.

Onink, V.: Model and analysis code for: "Empirical Lagrangian parametrization for wind-driven mixing of buoyant particulates at the ocean surface", https://doi.org/10.5281/zenodo.5764763, https://zenodo.org/record/5764763, 2021.

Onink, V., Wichmann, D., Delandmeter, P., and van Sebille, E.: The role of Ekman currents, geostrophy, and stokes drift in the accumulation of floating microplastic, Journal of Geophysical Research: Oceans, 124, 1474–1490, 2019.

Onink, V., Jongedijk, C. E., Hoffman, M. J., van Sebille, E., and Laufkötter, C.: Global simulations of marine plastic transport show plastic trapping in coastal zones, Environmental Research Letters, 16, 064 053, 2021.

Paris, C. B., Atema, J., Irisson, J.-O., Kingsford, M., Gerlach, G., and Guigand, C. M.: Reef odor: a wake up call for navigation in reef fish larvae, PloS one, 8, e72 808, 2013.

Pieper, C., Martins, A., Zettler, E., Loureiro, C. M., Onink, V., Heikkilä, A., Epinoux, A., Edson, E., Donnarumma, V., de Vogel, F., et al.: INTO THE MED: Searching for Microplastics from Space to Deep-Sea, in: International Conference on Microplastic Pollution in the Mediterranean Sea, pp. 129–138, Springer, 2019.

Poulain, M.: Etude de la distribution verticale de particules plastiques dans l'océan : caractérisation, modélisation et comparaison avec des observations, Ph.D. thesis, Institut National Polytechnique de Toulouse, 6 allée Emile Monso - BP 34038 31029 Toulouse, 2020.

Poulain, M., Mercier, M. J., Brach, L., Martignac, M., Routaboul, C., Perez, E., Desjean, M. C., and Ter Halle, A.: Small microplastics as a main contributor to plastic mass balance in the North Atlantic Subtropical Gyre, Environmental science & technology, 53, 1157–1164, 2019.

Riisgård, H. U. and Larsen, P. S.: Viscosity of seawater controls beat frequency of water-pumping cilia and filtration rate of mussels Mytilus edulis, Marine Ecology Progress Series, 343, 141–150, 2007.

Ross, O. N. and Sharples, J.: Recipe for 1-D Lagrangian particle tracking models in space-varying diffusivity, Limnology and Oceanography: Methods, 2, 289–302, 2004.

Samaras, A. G., De Dominicis, M., Archetti, R., Lamberti, A., and Pinardi, N.: Towards improving the representation of beaching in oil spill models: A case study, Marine pollution bulletin, 88, 91–101, 2014.

Taylor, J. R.: Accumulation and subduction of buoyant material at submesoscale fronts, Journal of Physical Oceanography, 48, 1233–1241, 2018.

Thompson, S. and Turner, J.: Mixing across an interface due to turbulence generated by an oscillating grid, Journal of Fluid Mechanics, 67, 349–368, 1975.

Van Sebille, E., Griffies, S. M., Abernathey, R., Adams, T. P., Berloff, P., Biastoch, A., Blanke, B., Chassignet, E. P., Cheng, Y., Cotter, C. J., et al.: Lagrangian ocean analysis: Fundamentals and practices, Ocean Modelling, 121, 49–75, 2018.

Van Sebille, E., Aliani, S., Law, K. L., Maximenko, N., Alsina, J. M., Bagaev, A., Bergmann, M., Chapron, B., Chubarenko, I., Cózar, A., et al.: The physical oceanography of the transport of floating marine debris, Environmental Research Letters, 15, 023 003, 2020.

Waterhouse, A. F., MacKinnon, J. A., Nash, J. D., Alford, M. H., Kunze, E., Simmons, H. L., Polzin, K. L., St. Laurent, L. C., Sun, O. M., Pinkel, R., et al.: Global patterns of diapycnal mixing from measurements of the turbulent dissipation rate, Journal of Physical Oceanography, 44, 1854–1872, 2014.

Wichmann, D., Delandmeter, P., and van Sebille, E.: Influence of near-surface currents on the global dispersal of marine microplastic, Journal of Geophysical Research: Oceans, 124, 6086–6096, 2019.

Yang, D., Chamecki, M., and Meneveau, C.: Inhibition of oil plume dilution in Langmuir ocean circulation, Geophysical Research Letters, 41, 1632–1638, 2014.

475  Zhao, D. and Li, M.: Dependence of wind stress across an air–sea interface on wave states, Journal of Oceanography, 75, 207–223, 2019.