# Peer review of "Empirical Lagrangian parametrization for wind-driven mixing of buoyant particles at the ocean surface"

_Geoscientific Model Development, 2021_

## Referee Comment (RC1)

**General comments**

The article submitted by V. Onink and collaborators and entitled **Empirical Lagrangian parametriza-tion for wind-driven mixing of buoyant particles at the ocean surface**, presents numerical results on the vertical motion of plastic particles induced by wind-driven mixing in a one-dimensional Lagrangian model of the ocean surface. The authors investigate two types of stochastic approaches to mimic the upper-ocean turbulent diffusion, as well as two different profiles of diffusion in the vertical based on published studies. They compare their numerical outputs, mainly the mean concentration profiles for plastic with different rising velocities, with observations from 5 previous studies (4 published, 1 unpublished).

The material presented here is well structured and clear, with the appropriate level of English. It corresponds to an interesting implementation of a Lagrangian transport model for plastic pollution based on models reproducing the properties of turbulence in the upper-ocean, and the authors indeed emphasized that their approach is compatible with more complex OGCM (Ocean Global Circulation Model) approaches. However, the discussion of the results made by the authors is limited to simple metrics. Furthermore, more efforts could be made in the description of the model implementation (although the code is available at a Zenodo deposit).

In the end, I have the impression that the results are not sufficiently discussed, and below are my main recommendations for the manuscript to be improved, before granting publication.

**1.** In §2.1, the code used for the study is described with little details. The code Parcel is clearly made for 2D or even 3D studies, but it is not clear to me how it is transformed to solve one-dimensional problems, in the vertical. What is the horizontal domain like, what is the rule of transport for the 100,000 particles transported all simultaneously launched at the same depth at the beginning? Much more details are required here. There are no details on the spatial resolution as well.

**2.** The comparison of the model outputs with the observations is made by using a single metrics, the root mean square error between mean profiles and a "normalized " field measurements. First a clear definition of the expression used is required although it might seems obvious, to avoid any confusion. Furthermore, it seems a bit too simplistic. Since the many profiles are not all with the same uncertainty, or the same flow conditions, some higher level of analysis could be made for the observations. Similarly, the temporal "steady" profile is not the only quantities to extract and variance at least would be of interest. Furthermore, the global comparison of a profile with observations by averaging with depth is possibly putting a lot of importance on strong errors at large concentrations although the overall profile could be 'on appearance' correct.

**3.** The case of the fastest rising particles is disappointing. The difficulties in terms of temporal resolution should be discussed in more depth, with some comments made on time intervals for fast objects related to the vertical resolution of the models too (0.03*30 1m ... to compare with vertical resolution). Furthermore, for the numerics to be relevant, some stronger recommendations in the conclusion should be made. To my mind, the modeling of such particles is not possible for current OGCM models unless a specific choice of temporal / spatial resolution is made, but I am not sure it is the correct interpretation to have here.

**Other comments**

Here is a list of other points of lesser importance.

- *l.78.* What is the value of alpha for $\delta t$ larger than $T_L$ (should be 0 I guess) ?

- *l.102.* The study is based on three sets of particles having different 'rise' velocities. It would be useful to discuss the values in comparison with the turbulent properties of flow (variance of $w'$ for instance).

- *l.140.* The introduction of $\theta$ is too succinct to be understood, more details like '$\theta$ is a Langmuir circulation enhancement factor that one can adjust between XX and YY, we choose $\theta = 1$ which corresponds to ..."

- *p5-6.* No reference in the text to Figure 1 for KPP profiles.

- *p7.* Table 1 introduces unpublished data which is almost invisible in the corresponding figures, and it represents a small number of profiles with little representation. Maybe it is not worth including them that way.

- *l.177.* Typo '$w_{10}$' instead of '$u_{10}$' ?

- *l.186.* (and at other lines too) The use of greater downward mixing is unclear. Discuss it in terms of depth, or larger number of particles at some depths, etc.

- *l.195.* 'With both KPP and SWB diffusion, M-1 models show increased leads to increased downward mixing of particles with increasing'. I am not sure I get this sentence clearly.

- *l.233-235.* The comment suggest that more plastic sampling in depth is needed, which is true, but I think they should also emphasize on the estimates of a proper diffusion model too (or of the eddy viscosity) !

- *l.238.* About the consistency of models. I understand the point by at the same time, why should it be consistent with other tracers if the model is inadequate? Plastics can also be a good indicator of a better diffusion model to be implemented, because it has a different nature (buoyancy, size, passive, etc). The reverse is of similar interest (test other model for tracers).

- *l.246-247.* One reference is missing for microplastic properties (Kooi. et al,... -¿ Poulain et al. 2018.

---

## Author Comment (AC1)

We'd like to thank both the reviewers for taking the time to read our paper submission and give their expert feedback. In the following document we have addressed their remarks and indicated where the specific changes to the document have been made. All line numbers are with regards to the tracked changes document.

**Referee: 1**

The article submitted by V. Onink and collaborators and entitled Empirical Lagrangian parametrization for wind-driven mixing of buoyant particles at the ocean surface, presents numerical results on the vertical motion of plastic particles induced by wind-driven mixing in a one-dimensional Lagrangian model of the ocean surface. The authors investigate two types of stochastic approaches to mimic the upper-ocean turbulent diffusion, as well as two different profiles of diffusion in the vertical based on published studies. They compare their numerical outputs, mainly the mean concentration profiles for plastic with different rising velocities, with observations from 5 previous studies (4 published, 1 unpublished).

The material presented here is well structured and clear, with the appropriate level of English. It corresponds to an interesting implementation of a Lagrangian transport model for plastic pollution based on models reproducing the properties of turbulence in the upper-ocean, and the authors indeed emphasized that their approach is compatible with more complex OGCM (Ocean Global Circulation Model) approaches.

We would like to thank the reviewer for these kind words.

However, the discussion of the results made by the authors is limited to simple metrics. Furthermore, more efforts could be made in the description of the model implementation (although the code is available at a Zenodo deposit).

In the end, I have the impression that the results are not sufficiently discussed, and below are my main recommendations for the manuscript to be improved, before granting publication.

1. In §2.1, the code used for the study is described with little details. The code Parcel is clearly made for 2D or even 3D studies, but it is not clear to me how it is transformed to solve onedimensional problems, in the vertical. What is the horizontal domain like, what is the rule of transport for the 100,000 particles transported all simultaneously launched at the same depth at the beginning? Much more details are required here. There are no details on the spatial resolution as well.

Parcels is indeed a python package that has been used for 1D, 2D and 3D Lagrangian studies, and we hope that our changes at lines 100 - 105 together with the code documentation at Zenodo give sufficient detail of the model setup. In short, we start with releasing 100,000 particles simultaneously at z=0, and we compute the vertical transport according to equations 3

- 7 depending on the scenario. We set the horizontal velocities to zero (thereby reducing the model to a 1D setup), and hence don't prescribe any horizontal domain. Finally, we calculate the $K_z$ fields with a vertical spatial resolution of 0.1 m, with the $K_z$ value at the particle position being linearly interpolated from these values.

2. The comparison of the model outputs with the observations is made by using a single metrics, the root mean square error between mean profiles and a "normalized " field measurements. First a clear definition of the expression used is required although it might seems obvious, to avoid any confusion. Furthermore, it seems a bit too simplistic. Since the many profiles are not all with the same uncertainty, or the same flow conditions, some higher level of analysis could be made for the observations. Similarly, the temporal "steady" profile is not the only quantities to extract and variance at least would be of interest. Furthermore, the global comparison of a profile with observations by averaging with depth is possibly putting a lot of importance on strong errors at large concentrations although the overall profile could be 'on appearance' correct.

We have added a definition of the RMSE at line 202 to clearly define the metric we use throughout the analysis. We acknowledge that the RMSE is not a perfect metric, but we struggled to find a better alternative. We looked at the correlation between the field data and model data (see the figure below), but we did not consider this a very informative metric because modeled decreasing concentrations with depth already provides a very high correlation; so the RMSE metric is much more stringent. Hence, we don't necessarily see the added value of including this metric.

We generally only have field concentrations, the wind and MLD (either precalculated or we could calculate it from provided CTD data) conditions at the time of sampling. Ideally we'd have wind speeds, MLD and langmuir mixing data for the hours leading up to the actual sampling to try and replicate the variability in ocean conditions, but as we state in lines 317 - 319, we are limited in the analysis we can do due to lacking such data, and we highlight in lines 325 - 327 and 336 - 338 that collecting more detailed information of sampled microplastics and the oceanographic conditions at the time of sampling would help further develop models such as ours. Nevertheless, we have expanded our analysis. We estimate the variability in the field measurements by binning the normalized field concentrations into 0.5 m bins and calculating the standard deviations over the binned data points for all wind conditions. While a large part of this variability is likely driven by time-varying ocean conditions or by e.g. different Langmuir mixing conditions under similar wind speed conditions (as stated in lines 315 - 323), it does provide a more robust estimate of the variability than solely plotting all field points. Due to the static wind and MLD conditions, the model variability is significantly smaller, but we do visualize this variability by shading around each profile, where the shading indicates the standard deviation at each depth level calculated over the final hour of each model simulation (Figures 2, 3, 5). We also better illustrate the time evolution of each profile in supplementary figure C1, which shows how quickly the model reaches an equilibrium assuming static wind and MLD conditions.

[Figure]

3. The case of the fastest rising particles is disappointing. The difficulties in terms of temporal resolution should be discussed in more depth, with some comments made on time intervals for fast objects related to the vertical resolution of the models too (0.03*30 1m ... to compare with vertical resolution). Furthermore, for the numerics to be relevant, some stronger recommendations in the conclusion should be made. To my mind, the modeling of such particles is not possible for current OGCM models unless a specific choice of temporal / spatial resolution is made, but I am not sure it is the correct interpretation to have here

We thank the reviewer for this insight. Indeed, the high buoyancy particles show strong sensitivity to the integration timestep with SWB diffusion, and we highlight this in the conclusions at lines 345 - 346: "The parametrizations generally perform well for timesteps of $\Delta t = 30$ seconds, but for high buoyancy particles users need to take care to use sufficiently short timesteps, especially with SWB diffusion". We also have expanded the discussion of the

influence that the boundary condition has on this integration timestep dependence in lines 270 - 271: "However, for $\Delta t = 30$ seconds the depth of mixing is now overestimated compared to smaller $\Delta t$ values (Fig. F2), as with $\Delta t = 30$ seconds and $w_r = 0.03$ m s$^{-1}$ the particle would be reflected up to 0.9 m below the ocean surface solely due to the model numerics. " In the case of the KPP diffusion this sensitivity doesn't appear to be as big an issue, as the near-surface $K_z$ are so small that even with small timesteps, the particle buoyancy dominates any mixing. As such, we feel that generally such high-buoyancy particles can be modelled within current OGCM models, but at least with these mixing parametrizations it appears such particles largely remain at the ocean surface, except in cases of especially strong mixing. As we state in line 274 - 276, it depends on the model application whether the error of $\approx$ 1m in the particle depths is acceptable, and whether shorter timesteps are computationally feasible: "Depending on the model application and setup, the error in the concentration profile depth ($\approx$ 1 m for high buoyancy particles) might be acceptable. Otherwise, the error can be reduced by using a smaller integration timestep where that is computationally feasible."

Other comments

Here is a list of other points of lesser importance.

- l.78. What is the value of alpha for $\delta t$ larger than TL (should be 0 I guess) ?

We have updated line 80 to state that we calculate alpha assuming dt <= TL. If dt were to be longer than TL, then the integration timestep would be too large to capture all relevant turbulent fluctuations, and a smaller timestep would be necessary.

- l.102. The study is based on three sets of particles having different 'rise' velocities. It would be useful to discuss the values in comparison with the turbulent properties of flow (variance of w 0 for instance).

We have added table A1 to show ratio of the rise velocity to the peak w' value for varying diffusion types and wind conditions, where w' is calculated with equation 3 for dt=30 seconds. We then briefly discuss this comparison of $w_r$ and w' in lines 111 - 114.

- l.140. The introduction of θ is too succinct to be understood, more details like 'θ is a Langmuir circulation enhancement factor that one can adjust between XX and YY, we choose θ = 1 which corresponds to ..."

Due to the feedback from reviewer 2, we have a much more extensive analysis on the influence of the Langmuir circulation enhancement factor $\theta$. As explained in lines 156 - 159, the presence of Langmuir circulation can significantly increase the amount of turbulent mixing within the mixed layer, and we investigate the influence of this by settings $\theta \in [1.0, 2.0, 3.0, 4.0, 5.0]$. As shown in figure 3, this can significantly increase the depth to which mixing occurs, and is an important process to consider when modelling vertical transport of buoyant microplastic (lines 226 - 241).

- p5-6. No reference in the text to Figure 1 for KPP profiles.

Fixed, with reference to the figure at line 154.

- p7. Table 1 introduces unpublished data which is almost invisible in the corresponding figures, and it represents a small number of profiles with little representation. Maybe it is not worth including them that way.

The vast majority of data points indeed originate from Kooi et al. (2016), but all these samples were collected within 5m of the ocean surface. Therefore, while smaller in number, the other profiles give us at least some insight of the vertical concentration profile for depths below 5m.

- l.177. Typo 'w10' instead of 'u10' ?

Indeed a typo, and now fixed.

- l.186. (and at other lines too) The use of greater downward mixing is unclear. Discuss it in terms of depth, or larger number of particles at some depths, etc.

Greater downward mixing can indeed be interpreted in various ways, and we've changed throughout the results and discussion to refer to deeper mixing, by which we mean that a greater number of particles is mixed deeper beneath the ocean surface.

- l.195. 'With both KPP and SWB diffusion, M-1 models show increased leads to increased downward mixing of particles with increasing'. I am not sure I get this sentence clearly.

The intended comment was that relative to M-0 models, using an M-1 results in more particles getting mixed deeper below the ocean surface, but the phrasing here did not communicate this clearly. We've rephrased this at line 243 to read "With both KPP and SWB diffusion, M-1 models show deeper mixing of particles as $\alpha \to 1$ (Fig. 5)."

- l.233-235. The comment suggest that more plastic sampling in depth is needed, which is true, but I think they should also emphasize on the estimates of a proper diffusion model too (or of the eddy viscosity) !

Indeed, we would benefit both from more field sampling of both plastics at depths and further measurements of near-surface mixing to validate mixing/eddy viscosity models. We now added to line 325 - 327 to emphasize this: "At the same time, we would encourage conducting more ocean field measurements of near-surface vertical eddy diffusion coefficient and/or eddy viscosity profiles, as this will allow further validation of the $K_z$ profiles predicted by the KPP and SWB theory with actual ocean near-surface mixing measurements."

- l.238. About the consistency of models. I understand the point by at the same time, why should it be consistent with other tracers if the model is inadequate? Plastics can

also be a good indicator of a better diffusion model to be implemented, because it has a different nature (buoyancy, size, passive, etc). The reverse is of similar interest (test other model for tracers).

That is a good point, and by comparing the modelled vertical concentration profiles with the field data, we show that Langmuir circulation mixing is likely a very important mixing process that needs to be accounted for when using KPP theory. Based on the new results with accounting for Langmuir mixing, we have rewritten this section now at lines 296 - 308 to read: "With regards to necessary data to calculate the diffusion profiles, the SWB approach has the benefit that it only requires surface wind stress data, while KPP diffusion additionally requires MLD data. Furthermore, our results indicate that accounting for LC-driven turbulent mixing improves KPP diffusion model performance, but determining which $\Theta$ value to use is not trivial. McWilliams & Sullivan (2000) demonstrated that $\Theta$ is inversely proportional to the Langmuir number La, which is defined as $La = \sqrt{u_{*w}/U_S}$ with $U_S$ as the surface Stokes drift. The Langmuir number can conceivably be calculated using OGCM data, but the details of such an implementation will be left for future work with 3D Lagrangian models. Furthermore, KPP diffusion has the advantage that it has been widely used and validated in various model setups (Boufadel et al., 2020; McWilliams & Sullivan, 2000; Large et al. 1994), while such extensive validation has not yet occured for SWB diffusion. Finally, the influence of wind forcing on turbulence is generally assumed to be limited to the surface mixed layer (Chamecki et al., 2019), while with the SWB profile wind-generated turbulence can extend below the MLD. To represent sub-MLD mixing, either a constant $K_z$ value or other $K_z$ profiles could be used, such as the $K_z$ estimates for internal tide mixing as proposed by (de Lavergne et al., 2020)." We then discuss how the results with comparing the modelled vertical concentration with the microplastic measurements allows indirect validation of the  KPP and SWB mixing estimates in lines 329 - 338: "The parameterizations have been validated for high/medium rise velocities, and at least for KPP diffusion with $\Theta>1.0$, the concentration profiles resemble those calculated from field observations. This provides confidence in the turbulence estimates from the KPP approach, and as these are independent of the type of particle that might be present, this would suggest the KPP approach can also be applied to neutral or negatively buoyant particles. However, as model verification was only possible for microplastic particulates with rise velocities approximately between 0.03 - 0.003 m $s^{-1}$, we would advise additional model verification for other particle types where the necessary field data is available. In the case of SWB diffusion, turbulent mixing seems underestimated when further from the ocean surface, and we would advise more validation with field observations before applying this diffusion approach to other particle types."

● l.246-247. One reference is missing for microplastic properties (Kooi. et al,... -¿ Poulain et al. 2018.

We have added references to Kooi et al. (2016) and Kukulka et al. (2012) at line 314, as these are the two studies we use field data from to validate our results that discussed how such vertical concentration profiles arise.

**Referee: 2**

The manuscript describes the vertical mixing of buoyant particles at the ocean surface, with comparisons made to microplastics field data. The model compares two different eddy diffusivity models, along with two types of Markov modelling (a random walk (M-0) and a higher order random walk which includes an autocorrelation timescale (M-1)).

The simulations do not represent a substantial contribution to modelling science.

We politely disagree with the reviewer and believe that our work is a substantial contribution to modelling science. We address the specific concerns about the diffusion approaches below, but in general there is a need for a near-surface wind mixing parametrization that can be applied to large-scale modelling efforts with OGCM data, and our work provides such a parametrization along with extensively documented model code that can be used as a basis for applying our parametrization to any given (Lagrangian) model setup. OGCM output, especially in the form of reanalysis products, generally does not provide turbulence fields as output, most likely for storage considerations. This severely hinders any 3D modelling studies of buoyant particles such as microplastics, as turbulent mixing is one of the main processes driving the form and depth of vertical concentration profiles; in this manuscript we provide an empirical workaround for this limitation. The reviewer has a number of concerns regarding the mixing/eddy viscosity models that we address below. These comments helped us to further develop our models, and this has resulted in relatively simple and computationally cheap parametrizations of near-surface mixing that capture the main features of vertical mixing, which we see as a substantial contribution to geoscientific model development.

One concern in this manuscript is the usage, discussion, and validation of the eddy viscosity models. The text describes two vertical diffusion models:

The first uses both the well-established Kukulka et al. (2012) for the near surface and extrapolates below using Poulain (2020). Poulain (2020) is a thesis and therefore the results therein have yet to be peer-reviewed. The Poulain (2020) experiment is described as a tank with a vertically oscillating grid. However, it is not clear in the text whether this model has been verified with respect to ocean surface mixing. Using more well-established near surface model would improve the model.

The near-surface parametrization from Kukulka et al. (2012) is indeed widely used to correct surface measurements of microplastic concentrations for vertical mixing, but Kukulka et al. (2012) itself emphasizes that the parametrization is only valid for depths up to approximately 1.5 times the significant wave height. As we now further highlight in lines 136 - 140, oscillating grid experiments have been widely used to study near surface turbulence, and have been shown to reproduce turbulence decay for velocities and dissipation rates that agree with measurements within the ocean surface mixed layer. Poulain (2020) is indeed a thesis, but the OGT experiments described within have been submitted for peer review, which at the time of writing is not finished yet. As such, we cited the thesis for the time being, and will update this to the peer-reviewed

article once it becomes available. However, we have been in contact with the lead author of the study, Dr. Marie Poulain-Zarcos, and she has confirmed that the eddy viscosity profile applied in our work has not changed during the peer-review process.

While direct validation of the eddy-viscosity model with ocean surface mixing has yet to occur, it does agree in general terms with Kukulka et al. (2012) in predicting constant mixing near the surface (within one significant wave height of the surface). The underlying theoretical reasoning behind the parametrization differs from KPP diffusion, and we think this provides an interesting contrasting approach for modelling near-surface mixing. Furthermore, our work provides an albeit indirect validation of the SWB diffusion approach, which seems to underestimate the total mixing throughout the mixed layer as we state in lines 336 - 338: "In the case of SWB diffusion, turbulent mixing seems underestimated when further from the ocean surface, and we would advise more validation with field observations before applying this diffusion approach to other particle types."

The second uses KPP. KPP is a bulk boundary layer model which goes to zero at the free surface. This means that all positively buoyant particles at the free surface would stay at the free surface, regardless of the wind conditions. The text mention this, but does not elaborate. Under this scenario, are their equilibrium profiles initial condition dependent? I would expect after long times, all the particles should stay at the surface, and therefore I'm unsure why simulations are needed when the final state is pre-determined.

We have rewritten lines 153 - 154 to clarify that the $K_z$ value at the surface in our formulation of KPP diffusion is not exactly zero: "As such, $K_z$ rises from a small non-zero value at z=0 to a maxima at z = 1 / 3 MLD, before dropping to $K_z=K_B$ for z $\leq$MLD (Fig. 1)." Furthermore, we have changed the x-axis in Figure 1 to a log axis, such that it is clear that while the near-surface $K_z$ is very small, it is not equal to zero. The reason for this is that we use a KPP formulation from Boufadel et al. (2020), which introduces a roughness scale of turbulence $z_0$, which can represent the surface roughness due to surface waves:

$$K_z = \left( \frac{\kappa u_{*w}}{\phi} \theta \right) (|z| + z_0) \left( 1 - \frac{|z|}{MLD} \right) + K_B$$

As such, even when z = 0, $K_z > 0$, and particles can get mixed down below the surface if the turbulence at z=0 is strong enough to overcome the rise velocity. For example, this can be seen in figure 2 for all particle types. In addition, in figure C1 we plot the time evolution of the medium buoyancy particles, which shows that for both SWB and KPP diffusion, the particles form an equilibrium vertical concentration profile due to the balance in buoyancy and turbulent mixing, where at a given time a significant number of particles are below the ocean surface. In changing ocean conditions, such a steady profile might not always emerge because during the time it can take to reach a steady profile (approximately 1- 2 hours according to our model results), the oceanographic conditions can change. However, this constant cycling of particles between different depths can for example affect the horizontal transport as the zonal and meridional ocean currents have been shown to vary with depth (for example, see Tsiaras et al., 2021). As

such, modelling the vertical mixing of buoyant particles is important to model the long-term fate of such particles, demonstrating how a parametrization such as ours can provide a substantial contribution to geoscientific model development.

Tsiaras, Kostas, et al. "Modeling the Pathways and Accumulation Patterns of Micro-and Macro-Plastics in the Mediterranean." *Frontiers in Marine Science* (2021): 1389.

Boufadel, Michel, et al. "Transport of oil droplets in the upper ocean: impact of the eddy diffusivity." *Journal of Geophysical Research: Oceans* 125.2 (2020): e2019JC015727.

The text then compares their simulation results to the observations, and they do not match. In fact, the simulations underpredict the vertical mixing of the microplastics. This is unsurprising, and has been previously demonstrated in the literature, where it has been noted that to fully account for proper vertical mixing of microplastics, one needs to include the effects of breaking waves and Langmuir turbulence (see e.g. Kukulka & Brunner 2015).

While the modelled vertical concentration profiles do show the decrease in concentration with depth as in the field observations, the M-0 models with both KPP and SWB diffusion seem to underpredict the depth to which particles are mixed. In our original approach, we followed Boufadel et al. (2020) in assuming that the influence of Langmuir circulation (LC) driven turbulence was negligible, which in equation 11 for the KPP $K_z$ profile corresponds to setting the Langmuir circulation enhancement factor $\theta = 1.0$. However, as the reviewer notes, studies such as Kukulka & Brunner (2015) and Brunner et al. (2015) have shown that LC-driven turbulence is necessary to properly account for the vertical mixing of microplastics, so we tested the influence $\theta$ has on the mixing of our buoyant particles. According to McWilliams & Sullivan (2000), LC-driven turbulence can increase mixing by $\theta = 3 - 4$, so we considered $\theta \in [1.0, 2.0, 3.0, 4.0, 5.0]$ as outlined on lines 156 - 159. As we now show in Figure 3, $\theta > 1.0$ increases the depth to which particles are mixed, and generally increases the agreement between the field observations and the modelled concentration profiles. We acknowledge on lines 299 - 303 that selecting the correct $\theta$ value for a simulation is not trivial, as McWilliams and Sullivan (2000) show that this is inversely proportional to the Langmuir number, which in turn can vary with time and space. However, we consider this to be a modelling choice that will depend on the larger 2D or 3D model setup within which our parametrization would be applied, so we shall leave that to future work for now. Within this paper, we consider it sufficient that we have now demonstrated that by correctly setting the value of $\theta$, we are able to more accurately predict the vertical mixing of buoyant particles.

We agree that it is a weakness of the KPP diffusion approach that KPP theory does not truly account for surface wave breaking, which can lead to significant mixing at the surface, as shown by Kukulka et al. (2012) and also our own SWB diffusion approach. Boufadel et al. (2020) suggested that the surface roughness $z_0$ could be used to account for surface wave breaking, and while this would require a lot more theoretical work with turbulence theory to prove or disclaim, the $z_0$ term does provide us with a simple way of testing the model sensitivity to high near-surface $K_z$ values (as we state in lines 168 - 170). In our original formulation, we set $z_0$ according to Zhou &

Li (2019), which leads to $z_0 = 2.38 \times 10^{-6} - 2.86 \times 10^{-4}$ m. We now consider an alternative formulation where the roughness is a fraction of the significant wave height $z_0=0.1Hs$, where we calculate the significant wave height for a given wind condition according to Kukulka et al. (2012). This has minimal impact on the magnitude of $K_z$ for depths greater than ~1 m, but does lead to higher $K_z$ as $z \to 0$ (Figure B1). We show in figures 3, D1 and D2 that including this higher near-surface mixing can increase the depth to which particles are mixed below the surface, but overall, the effect is smaller than that of LC-driven mixing through $\theta$. This agrees with the conclusions of Brunner et al. (2015) that LC turbulence has a stronger effect on the overall vertical concentration profiles than surface wave breaking does. As such, we conclude on lines 285 - 288 that: "although we recommend future work incorporating surface wave breaking into KPP theory, our current KPP diffusion approach representing LC-driving mixing through $\theta$ seems to capture the majority of turbulent mixing dynamics."

Kukulka, T., and K. Brunner. "Passive buoyant tracers in the ocean surface boundary layer: 1. Influence of equilibrium wind-waves on vertical distributions." *Journal of Geophysical Research: Oceans* 120.5 (2015): 3837-3858.

Brunner, K., et al. "Passive buoyant tracers in the ocean surface boundary layer: 2. Observations and simulations of microplastic marine debris." *Journal of Geophysical Research: Oceans* 120.11 (2015): 7559-7573.

McWilliams, James C., and Peter P. Sullivan. "Vertical mixing by Langmuir circulations." *Spill Science & Technology Bulletin* 6.3-4 (2000): 225-237.

Zhao, Dongliang, and Moxin Li. "Dependence of wind stress across an air–sea interface on wave states." *Journal of Oceanography* 75.3 (2019): 207-223.

Overall, this study does not add any new contributions to the field. The eddy diffusion profiles do not advance the state of the modelling, as they both have clear faults, and because they do not include all the relevant physics needed to fully explain the observations (Langmuir turbulence and/or breaking waves), it is hard to draw any conclusions from their comparisons to the data.

Vertical mixing can have a significant impact on the ultimate fate of buoyant particles in the ocean, and given that turbulence data from OGCMs is not readily available, we are convinced that our parametrizations provide a useful contribution to the modelling field. We would like to thank the reviewer for raising concerns about including all relevant physical processes, as we originally underestimated the influence of especially LC-driven mixing. We have now shown that by setting the Langmuir circulation enhancement term $\theta > 1.0$, our model can more accurately predict the depth to which buoyant particles are mixed with KPP diffusion. Properly accounting for surface wave breaking remains a weakness of KPP theory as a whole, but we have shown that the influence of such elevated near-surface $K_z$ values on the overall vertical concentration profile is not as significant as with LC-driven mixing, such was also shown by Brunner et al. (2015). Therefore, we feel that our parametrizations do provide a useful, new contribution to the field, with our own documented model code being available to act as a foundation for any application of our parametrization to a larger 2D or 3D model setup.

[revised manuscript text omitted]

---

## Referee Report (RR1)

**General comments**

The revised version of the article submitted by V. Onink and collaborators and entitled **Empirical Lagrangian parametrization for wind-driven mixing of buoyant particles at the ocean surface**, has replied to all the recommendations made by the two reviewers. To my mind, efforts for a more complete discussion of their results have been made. Nevertheless, by reading the new material provided in the revised version, below are some recommendations that need to be addressed for the manuscript to be granted publication.

1. By discussing the KPP model in greater details, especially by considering the influence of the parameters  $\theta$  describing the Langmuir Turbulence (LT), the authors have improved the quality of the comparison with observations. However, the SWB model also has adjustable parameters, the most obvious one is the depth over which the intensity of turbulence is constant. The choice to make the transition for the decay of turbulence at  $H_s$  is as arbitrary as the value for  $\theta$  to represent the intensity of the LT. Thus, it should be associated with a parametric study as well because changing this value into  $1.5H_s$  or  $2H_s$  could lead to improved comparison with observations as well. The sake of a parametric study for SWB is also to give an equivalent attention to the two models, there is currently a stronger emphasis on the KPP model. If the outputs are revisited for SWB, it can also modify the conclusion that KPP model performs better with respect to observations.

2. The goal of the comparison of the two diffusion models (KPP vs SWB) is to discuss the influence of the physics important for vertical transport modeling, and no model alone does a perfect job, although KPP with strong enough LT seems a better choice. The ultimate question that should be considered in this context is the question of adding up the two models.

**Other comments**

Here is a list of other points of lesser importance.

- *l.271.* 'by' instead of 'be' ?
- *l.319-320.* The variance in the modeled data is much less here because the numerical runs are 1D, and does not reproduce the fluctuation of ocean dynamics... It is unlikely that wind condition are the only origin of this variability in observations (currents, fronts, meso-scale eddies, etc).
- *l.457-459.* The reference Poulain et al. 2018 is with the wrong year. The online version of the paper is 2018, the official (doi) reference is in 2019 (53(3), 1157-1154).

---

## Author Response (AR2)

We'd like to thank the reviewer for taking the time to read our paper submission and give his expert feedback. In the following document we have addressed his remarks and indicated where the specific changes to the document have been made. All line numbers are with regards to the tracked changes document.

**General comments**

The revised version of the article submitted by V. Onink and collaborators and entitled Empirical Lagrangian parametrization for wind-driven mixing of buoyant particles at the ocean surface, has replied to all the recommendations made by the two reviewers. To my mind, efforts for a more complete discussion of their results have been made. Nevertheless, by reading the new material provided in the revised version, below are some recommendations that need to be addressed for the manuscript to be granted publication.

1. By discussing the KPP model in greater details, especially by considering the influence of the parameters  $\theta$  describing the Langmuir Turbulence (LT), the authors have improved the quality of the comparison with observations. However, the SWB model also has adjustable parameters, the most obvious one is the depth over which the intensity of turbulence is constant. The choice to make the transition for the decay of turbulence at Hs is as arbitrary as the value for  $\theta$  to represent the intensity of the LT. Thus, it should be associated with a parametric study as well because changing this value into 1.5Hs or 2Hs could lead to improved comparison with observations as well. The sake of a parametric study for SWB is also to give an equivalent attention to the two models, there is currently a stronger emphasis on the KPP model. If the outputs are revisited for SWB, it can also modify the conclusion that KPP model performs better with respect to observations.

We thank the reviewer for raising this point, and we have made a number of revisions throughout the text to better balance the emphasis given to the two diffusion approaches. Specifically, we have introduced a new parameter  $\gamma$ , which controls the depth to which we have constant mixing as a multiple of the significant wave height. As we state in lines 144 - 145, there is uncertainty in what value  $\gamma$  should take, as Poulain et al. (2020) implies  $\gamma = 1.0$  while based on Kukulka et al. (2012) it would be  $\gamma \approx 1.5$ . As such, we now consider  $\gamma \in [0.5, 1.0, 1.5, 2.0]$ . As shown in the new figures 4, E1 and E2, taking higher  $\gamma$  values results in deeper mixing of all particle types, and leads to better agreement with the field measurements (Figure 5). Overall, KPP diffusion can still lead to deeper particle mixing, but as we do not have sufficient field data below 5m we now state in line 271: "Considering the KPP and SWB diffusion profiles, the results in this study are inconclusive with regards to which approach performs better relative to field observations." 2. The goal of the comparison of the two diffusion models (KPP vs SWB) is to discuss the influence of the physics important for vertical transport modeling, and no model alone does a perfect job, although KPP with strong enough LT seems a better choice. The ultimate question that should be considered in this context is the question of adding up the two models.

Although using  $\gamma = 1.5 - 2.0$  does improve the model performance of SWB diffusion relative to the field observations, we still consider KPP with sufficiently strong LT mixing a better parametrization choice (although as we outline in lines 292 - 297, setting the appropriate  $\theta$  value is not trivial). We considered the possibility of combining the two diffusion approaches in some fashion, but ultimately, we concluded that expanding the KPP diffusion approach in a theoretical fashion would be beyond the scope of this study. Simply adding up the two diffusion models is a possibility, but this would mean that there is no longer one consistent theoretical framework underlying the parametrization. Furthermore, it would also imply that the wind-driven mixing is no longer constrained by the MLD, which is an important feature of the KPP diffusion model.

To examine the influence of increased near-surface  $K_z$  values, we did include the KPP model modification where we set the roughness scale  $z_0 = 0.1 x H_s$ . This led to higher near-surface  $K_z$  values, but as we state in lines 306 - 311, overall this was a much weaker influence on the overall concentration profile than LT turbulence (as similarly shown by Brunner et al., 2015). As such, while we acknowledge that the KPP diffusion approach is not a complete representation of all turbulence processes within the surface mixed layer, it does capture the majority of turbulent mixing dynamics. This makes it suitable to be applied to model vertical particle transport in a larger 3D model setup.

**Other comments**

- L. 271: "by" instead of "be"? Fixed.

- L. 319-320. The variance in the modeled data is much less here because the numerical runs are 1D, and does not reproduce the fluctuation of ocean dynamics... It is unlikely that wind condition are the only origin of this variability in observations (currents, fronts, meso-scale eddies, etc).

Indeed, there are many different oceanographic processes that contribute to the variance observed in the field measurements, and we intended our mention of wind to be an example, rather than suggesting it is the only relevant process. As such, we have updated the text at lines 323 - 325 to clarify this: "This is in part also due to assuming constant environmental conditions

over 12 hours for the model simulations, while wind and other oceanographic conditions can change on much shorter timescales over the ocean surface."

- The reference Poulain et al. 2018 is with the wrong year. The online version of the paper is 2018, the official (doi) reference is in 2019 (53(3), 1157-1154)

Fixed.